


# Resilience issues and challenges into built environments: a review

Charlotte Heinzlef [1], Bruno Barroca [2], Mattia Leone [3], Thomas Glade [4], Damien Serre [1]

[1] UNIV. POLYNESIE FRANCAISE, IFREMER, ILM, IRD, EIO UMR 241, Tahiti, Polynesie Francaise , charlotte.heinzlef@upf.pf, damien.serre@upf.pf

[2] Lab'Urba, Université Gustave Eiffel - 16 Boulevard Newton, 77420 Champs-sur-Marne, France, bruno.barroca@u-pem.fr

[3] University of Naples Federico II, Department of Architecture - Via Toledo 402, 80134 Napoli, Italy, mattia.leone@unina.it

[4] Department of Geography and Regional Research, University of Vienna - Universitaetsstr. 7, A-1010 Vienna , Austria, thomas.glade@univie.ac.at

## Abstract

This paper proposes a review of existing strategies and tools aiming at facilitating the operationalization of the concept of resilience into built environments. In a context of climate change, increased risks in urban areas and growing uncertainties, urban managers are forced to innovate in order to design appropriate risk management strategies. Among these strategies, making cities resilient has become an imperative. This injunction to innovation fits perfectly with the urban, economic, political, social and ecological complexity of the contemporary world. As a result, the concept of resilience is integrated into the issues of urban sprawl and the associated risks. However, despite this theoretical and conceptual adequacy, resilience remains complex to integrate into the practices of urban planners and territorial actors. Its multitude of definitions and approaches has contributed to its abstraction and lack of operationalization. This review highlights the multitude of approaches and methodologies to address the bias of the lack of integration of the concept of resilience in risk management. The limit is the multiplication of these strategies which lead to conceptual vagueness and a lack of tangible application at the level of local actors. The challenge would then be to design a toolbox to concentrate the various existing tools, conceptual models and decision support systems in order to facilitate the autonomy and responsibility of local stakeholders in integrating the concept of resilience into risk management strategies.

## 1. Introduction: several disciplines, definitions and associated concepts

Operationalizing resilience is a complex, even conflicting subject. Because of its multidisciplinary origin and the multitude of approaches, interpretations of resilience and its operationalization are sometimes contradictory (Davoudi et al., 2012). This contradiction is essentially due to the fact that resilience belongs to many disciplines, physics, psychology, ecology or risk management. This disciplinary and conceptual vagueness makes the use of resilience and its integration into risks . The concept of resilience is faced with a problem of formalization which makes it difficult to move from theory to practice (Weichselgartner and Kelman, 2015). Despite its growing success, the operational relevance of the concept is therefore constantly questioned and questioned .

### 1.1. Resilience, at the crossroads of several disciplines

Over the past 20 years or so, resilience has become an integral part of risk management (Heinzlef et al., 2020a). However, its multidisciplinary use makes it a polysemic and abstract concept (Bahadur et al., 2010). This concept is today over-used, solicited in many fields and linked to many notions (Emrich and Tobin, 2018). From the Latin *resalire* (*re*, backwards; *salire,* jump), the term resilience is used for the first time to illustrate the idea of 'bouncing' to refer to the noise that the echo makes while 'bouncing'. The first meaning of the word resilience in the English language therefore means "to bounce" (Saunders and Becker, 2015), "to straighten up". In French, the meaning of the word evolved during the Middle Ages by taking on the meaning of to retract, to free oneself from a contract by a kind of jump backwards. Nevertheless, this is the meaning of the Anglo-Saxon term that persists today linked to qualities of elasticity, springiness, resourcefulness. The Latin root indicates quite clearly the interpretation of the term: the capacity to untie/mitigate the impacts of a trauma. However, in view of the many definitions


(Hosseini et al., 2016) and fields of use, it would be more accurate to begin by talking about resiliences
rather than resilience (Emrich and Tobin, 2018). This multitude of "resiliencies" (Bec et al., 2016) can
be explained by its origins but also by its diverse and varied uses (Gaillard, 2007; Klein et al., 2003).
Each actor can define the term resilience in different ways (Meerow et al., 2016). This diversity of
interpretation also makes it a weakness, which explains why translating this concept into action
strategies is difficult and laborious. From an innovative concept to a buzzword, resilience is a source of
enrichment, learning and improvement as an abstract word that few actors understand and integrate into
their risk management strategies. This is why it is necessary to understand the different definitions, and
therefore interpretations, that are related to this concept. While interdisciplinary can serve and enrich
the understanding of resilience, it can also serve it in its transition to operationalization.
1.1.1.Concept of resilience in physics
The first use of the concept of resilience in science is in the field of physics. In Thomas Tredgold's
Practical Treatise on the Strength of Cast Iron and Other Metals (1824), resilience refers to the elasticity
and strength of materials. In particular, it refers to the ratio of the absorbed kinetic energy required to
cause the rupture of a metal and therefore to the capacity of the metal to resist the impact while keeping
its initial shape (Campbell, 2008). Following a continuous pressure of a material under the effect of a
stress, the return to its initial state is the phenomenon of physical resilience. Resilience is therefore an
intrinsic - measurable - capacity.
1.1.2.Resilience in psychology
Psychological resilience is defined as an individual's ability to adapt in the face of tragedy, trauma,
disruption, threats or stress (Booth and Neill, 2017). The idea is to move beyond the difficult situation
(Cyrulnik and Jorland, 2012). Several approaches have followed, some defining psychological resilience
as a personal quality, others as a process, or as an ability, strength or aspiration that each individual
possesses. Today's established definition is that resilience represents positive adaptation in the face of
adversity (Luthar et al., 2000). Adaptation can be defined as significant and/or positive depending on
the situation (Luthar, 2015). In this case, resilience is not measured directly but is inferred from the
actions of individuals and evidence of adaptation. In this approach, risk or stress is required to
demonstrate resilience. Resilience is therefore distinct from normal development, i.e., undisturbed
individual development (Rutter, 1999, 1987; Rutter and Zigler, 2000).
1.1.3.Resilience concept in ecology
In 1973, Holling defined resilience as the ability of an eco "*system to maintain its qualitative*
*structure*" (Holling, 1973). This definition emphasizes the capacity of a system to maintain its qualitative
structure (Holling, 1973), to absorb a shock without changing behavior, function. It is therefore above
all the notion of persistence that is put forward. The idea is that the system has a constant evolution,
characterized by pendulum movements towards the initial state preceding the disturbance. However, the
idea that there is a single initial state of equilibrium for any element has been widely criticized, especially
when analyzing complex systems characterized by their evolution. This is why, several years later,
Holling evolved by introducing the idea of evolution without relying on necessarily on a return to a pre-
existing equilibrium. The resilience of an ecosystem can therefore be defined as the capacity to absorb
disturbances while reorganizing itself (Walker et al., 2004) in a feedback process. Gunderson and
Holling (Chelleri, 2012; Gunderson and Holling, 2002) have therefore innovated by using the panarchic
concept to illustrate the dynamics and multi-scale dimension of resilience. The Panarchy model sets up
a dynamic cycle combining a growth phase (exploitation phase), conservation (equilibrium phase),
collapse (release phase) and finally a reorganization phase.
1.1.4.Resilience at the crossroads of disciplines for new risk management
The Hurricane Katrina disaster in New Orleans in 2005 marked a major turning point in the
history of the turning point in risk management (Campanella, 2006; Cutter et al., 2008a, Hernandez,


2009). To prevent a similar event from happening again, risk management has evolved to incorporate the concept of resilience. The objective is to use this concept to best prepare populations and territories to increased risks in urban areas. The idea is no longer to analyze the risks in a compartmentalized manner but to study the disruptive event and its consequences as a whole. Three approaches and methods stand out currently (Folke, 2006; Folke et al., 2010, 2002):

- The engineering approach assumes a steady state (Brand and Jax, 2007; Holling, 1973). The idea is to evaluate the gap between the disturbed state and the steady state and the speed of return. to a state of equilibrium after a disturbance. The hazard here represents an element against which you have to protect yourself and avoid.
- The ecosystem approach (Carpenter et al., 2001; Gunderson and Holling, 2002; Walker et al., 2004) does not imply a return to a previous equilibrium but acknowledges that several states of equilibrium.
- The socio-ecological approach is defined as "*the capacity of a system to absorb disruptions and to reorganize while undergoing change, so as to still maintain its overall function, structure, and feedback loops, and by identity; in other words, the ability to change in order to maintain the same identity. identity*" (Folke et al., 2010). This approach differs from the other two because, while it integrates the idea of absorbing disturbances, it also incorporates the notions of learning, adaptation, self-organization.

The multiple disciplinary origins of the concept of resilience make it difficult to define. There are many meanings behind this disciplinary identity, creating a lack of understanding between scientific experts and/or local actors.

### *1.2. Attempted resilience definitions*

While resilience belongs to so many different disciplines, there are many definitions related to it. The main idea, however, is that when faced with a shock, a crisis, the system (whatever it is) disappears or recovers. However, the question remains: when can it be determined that a system has recovered from how many disturbances, changes and transformations it has undergone?

The various definitions belonging to these various disciplines refer to concepts such as: recovery, reconstruction, restoration, renewal, return to a state of equilibrium, return to a previous state, rebound, etc.

These different points of view then refer to resilience according to two currents of thought:

- Resilience is either a process or the result of this process: we evaluate the resilience capacity of a post-crisis system (result), or the succession of solutions developed by this system to recover from a shock (process). This vision of resilience can therefore only be assessed *a posteriori*, in order to evaluate whether the system has been able to maintain itself beyond a shock and to overcome it (Reghezza-Zitt and Rufat, 2016).
- Resilience is an intrinsic capacity of the system, a capacity that can be put forward at the time of the shock. It can then be translated as an ability, a capacity, or even a capability. This resilience can therefore be pre-existing to the shock, innate or acquired. This resilience capacity can be declined according to several characteristics:
  - Resistance capacity: Serre (2018) defined three capacities of resilience and defined the resistance ability to determine "the physical damage to the network as a result of the hazard" (Serre et al., 2013). It is essential to know before any risk management and actions plan the potential damages of a system, in order to adapt resilience strategy. It is estimated that, more the technical system is damaged, greater is the possibility of a malfunction of the system and more it will be difficult to restore it to service.
  - Absorption capacity: For instance, the UNISDR (2009) has define resilience as the "*ability of a system, community or society exposed to hazards to resist, absorb, accommodate to and recover from the effects of a hazard in a timely and efficient manner, including through the preservation and restoration of its basic structures*


*and functions*". Cardona (2004) defined resilience as the capacity of the damaged ecosystem or community to absorb negative impacts and recover from these.

- o Adaptive capacity: Pelling (2011) defends the idea that resilience is the ability of an actor to cope with and adapt to hazards stress. . It refers to the "*ability of systems, institutions, humans and other organisms to adjust to potential damage, to take advantage of opportunities, or to respond to consequences*" (IPCC, 2014). This implies considering the entire pool of assets (social, physical, financial, natural, human, and cultural) and resources (technological, knowledge and governance) which can be mobilized to build resilience to climate change impacts. Socio-technical end ecological aspects are equally targeted in a systemic perspective (Whitney et al., 2017), including consideration of trade-offs among them to avoid social-ecological traps which can risk conditions (Carpenter and Brock 2008).
- o Reaction capacity, linked to self-organization: Pickett et al. (2004) have defined resilience as the "*ability of a system to adjust in the face of changing conditions*" and Ahern (2011) has defend resilience as a "*capacity of systems to reorganize and recover from change and disturbance*".
- o Ability to rebuild using internal and external forces: Walker et al. (2004) developed the idea that resilience is the capacity to "*reorganize while undergoing change so as to still retain essentially the same function, structure identity, and feedbacks*"
- o Learning capacity: The Resilience Alliance (Walker and Salt, 2006) defends that resilience is a combination of three capacities, absorb and remain within the same state, the capacity of self-organization and "*the degree to which the system can build and increase the capacity for learning and adaptation*" (Carpenter et al., 2001; Klein et al., 2003)
- o Ability to bounce back or reach a new state of equilibrium: to some authors, there is one single-state equilibrium which implies to bounce back to equilibrium previous disturbance (Holling, 1996). On the contrary, others consider that we can observe multiple-state equilibrium which suppose that systems have different stable states (Davoudi et al., 2012; Holling, 1996)

These different capacities can be self-sustaining or, on the contrary, contradict each other (such as the capacities of resistance and adaptation). Faced with these different positions, the notions and concepts associated with that of resilience accentuate the abstraction and incomprehension of the concept.

### *1.3. Concepts associated to resilience perception*

No doubt a victim of its multitude of disciplines and definitions, resilience has been continually associated with or compared to related concepts. Resilience is regularly compared or associated with the concepts of vulnerability and sustainable development (Romero-Lankao et al., 2016).

#### 1.3.1. Resilience vs Vulnerability

The classic way of analyzing resilience and vulnerability is to contrast them: if you are resilient, you are not vulnerable and vice versa (Folke et al., 2002). This clear opposition seems logical: if resilience is the ability to adapt to a shock and vulnerability is defined as the propensity to damage, then the more vulnerable a concept is, the less resilient it is (Pelling, 2003). So the equation is simple, reducing vulnerability is the same as increasing resilience (Klein et al., 2003).

Yet this opposition has been widely contested. First of all, social vulnerability reflects the capacity to face, anticipate and adapt to risks .These social capacities are largely integrated into the notion of resilience (Cardona, 2004). Resilience can therefore be seen as an integral part of the concept of vulnerability (Britton and Clark, 2000), being aimed "*to not only restore functionality but also correct existing social, political, and economic structures that may have increased exposure and constrained capacity to cope with the crisis*" (Patel and Nosal, 2016). Thus, the two concepts cannot be completely opposed. Concerning the positioning aimed at qualifying the concept of vulnerability as "negative",



"positive" vulnerability provides a counter-argument. Vulnerability is considered positive when it leads
to a change that brings about a beneficial transformation (Gallopín, 2003) .For example, in a situation
of vulnerability in a dictatorial political system, its collapse is positive. Seeing the collapse or paralysis
of an urban system following a flood can raise awareness and allow it to evolve towards more
appropriate functioning. Indeed, while in risk assessment vulnerability is in general hazard-specific,
certain factors - such as poverty, the lack of social networks and social support mechanisms, inadequate
governance structures - will aggravate or affect vulnerability levels irrespective of the type of hazard
(Prowse, 2003; UNEP, 2003). Such dimensions of resilience, which involve society and ecosystems as
a whole, can be used to identify cross-cutting vulnerability aspects to be tackled as high-level policy and
governance issues, linked e.g. to limitations in access to and mobilization of the resources of individuals
and institutions, as well as to the incapacity to anticipate, adapt, and respond to absorb the socio-
ecological and economic impact of hazards (Miller et al., 2010; UNISDR, 2011; Cardona et al., 2012;
EEA, 2016). Under these conditions, vulnerability and resilience are no longer in opposition but are part
of a whole. They can then be approached along a continuum. This new stance leads to the notion of
resilient vulnerability (Provitolo, 2012). This notion reflects the idea that "*vulnerability can be traversed*
*and modified by resilience considered from a global perspective, i.e. that this resilience can, on the one*
*hand, be directly linked to the vulnerability to which it applies and, on the other hand, have a positive*
*or negative effect depending on the scale at which the system is studied*" (Provitolo, 2012).
In conclusion, resilience and vulnerability are not dichotomously opposed. The two concepts are
equally adaptable to technical and/or social systems. Resilience and vulnerabilities overlap in their
approach to systems to provide a vision of exhaustive of the elements composing this one. Addressing
the two concepts leads to an analysis of the question of long-term risks. It is therefore necessary to learn
to live with the change and uncertainty and not seek short-term control of risks. Analyze together the
two concepts is like learning from crises (vulnerability approach) and innovate to adapt to risks
(resilience).
243         1.3.2.Resilience vs Sustainable development
Faced with increasing risks, stakeholders have identified two concepts (Saunders and Becker, 2015),
that of resilience (taking into account the management of disturbances) and that of sustainable
development (analyzing the balanced economic, social and environmental development of the territory).
For some, resilience is a necessary condition for sustainability (Folke et al., 2002; Klein et al., 2003).
For others, after studying the possible trajectories of ecosystems according to different initial states,
resilience is not sufficient, sometimes it is not even necessary.
However, Toubin et al. (2015) defend the fact that resilience can play a role in the realization of the
sustainable city (Elmqvist et al., 2019), an ideally functioning urban system. The urban resilience
enhancement approach is then defined as a means of managing the jolts of the urban system subjected
to numerous disturbances (short-time resilience) and maintaining it in the ideal trajectory of
sustainability (long-term resilience) linked to a system state indicator (economic growth, carbon
footprint, or demographics, for example). Resilience is thus presented as a means of achieving
sustainability (Toubin et al., 2015). Nevertheless, resilience may also "*run counter to sustainability*
*goals: for instance, efficiency reduces diversity and redundancy, both of which are key features of*
*resilience. This conflict is illustrated by high-density urban areas, which can be more efficient to run in*
*terms of, say, energy distribution, communications and waste collection. However, these areas can also*
*be vulnerable to extreme events such as flooding because they are less diverse (with few green areas,*
*for example) and have few redundancies (in the form of back-up facilities and disaster-management*
*processes)*" (Elmqvist, 2017).
Resilience as the capacity of a system to adapt to disturbances thus appears better able to satisfy the
need to operationalize the sustainable city. Indeed, the normative basis of sustainable development,
particularly in the expression of the major global principles, "freezes" the ideal model to be achieved,
while its subjective character raises many debates as to the - moral - values to be pursued. Conversely,
resilience seeks to free itself from norms in favor of descriptive magnitudes and ensure a better reactivity
of the urban system in the face of the unexpected. "*Improving resilience increases the chances of*
*sustainable development in a changing environment where the future is unpredictable and surprise is*





*likely*." (Folke et al., 2002). Developing a sustainable territory and community cannot therefore be
envisaged without a long-term resilience strategy.
The concept of resilience is a multifaceted concept, involving a plurality of disciplines, definitions,
notions and associated concepts. This diversity can be interpreted both as a source of opportunity but
also as a difficulty in the operationalization of resilience and its lack of integration into risk management
strategies. In the face of new risks linked to climate change, the evolution of urban areas and the
concentration of issues (part 1), the concept of urban resilience represents both an innovative and
essential concept but also full of operational limits both at the international and local levels (part 2).
This is why a variety of methods, concepts and strategies have been developed to address the issue of
operationalization and appropriation of the concept by local actors in order to respond to these limits of
application (part 3). We will conclude on the notable advances in these approaches to integrate the
concept of resilience by presenting the next steps needed to respond to the limits still present in the
scientific and operational field.
**2. Urban risks: over-urbanization, cascading effects and multi-risk approach**

The current climate change context has led to an increase in natural disasters of about 2% per year
for the past 15 years (Catastrophes Naturelles-Observatoire permanent des catastrophes naturelles et des
risques naturels, 2016). At the same time, the increase in the number of people and goods in urban areas
is making it more fragile. considerably the cities. Today, nearly three out of five cities, with 500,000
inhabitants, are at risk. However, urban areas produce between 70 and 80% of the world economy and
are home to 55% of the world's population , with an increasing urban-rural drift expected to raise this
value up to 68% by 2050 (UNDESA, 2019; Zevenbergen et al., 2010). Such a concentration of stakes
increases the impact of disasters and raises questions on the future of cities.
*2.1. Over-urbanization*
In 2008, half of the world's population lived in urban areas. This concentration is likely to accelerate.
Projections show that urbanization, combined with overall world population growth, could add an
additional 2.5 billion people to urban areas by 2050 (United Nations, 2018). The unplanned expansion
of urban areas to face to this rapid growth, combined with inappropriate land-use planning, a
geographical location at risk (river mouth, swampy areas, major river bed, etc.) and difficult regulation
of building standards, contributes to the over-vulnerability of urban territories and populations. Urban
areas in coastal regions are particularly exposed to sea level rise. Low-lying coastal areas - less than 10
metres above sea level - account for just 2% of the world's land but are home to 13% of the world's
urban population. In 2007, Africa had 37 cities with more than 1 million inhabitants, half of which are
located - at least in part - in the low-lying coastal zone.
However, this tendency to focus on a specific area can be observed on a global scale: cities occupy
only 1% of the world's territory (Angel et al., 2018) . Developed countries have therefore never
concentrated more value added per km$^2$ than they do at present. This concentration of population on
such a small portion of the territory has increased spatial and social vulnerability through the exposure
of the issues. Indeed, it seems logical to consider that the more a population and its issues are
concentrated in a small area, the greater the damage will be. Flooding in an uninhabited area will not be
considered and apprehended in the same way as in a metropolis (Mitchell, 1999).
The example of storm Xynthia in France in 2010 is an example of the effects of over-urbanization
on the reality of the disaster. This storm is one of the deadliest disasters in France with 59 deaths. The
marine submersion, which reached 1.53 meters in La Rochelle, affected some communes up to 85% of
their surface area (Duvat, 2011). The magnitude of the disaster was due in particular to demographic




change and rapid urbanization in the area. Thus, between 1946 and 2007, urbanization in the lower areas
doubled or even tripled in some communes, leading to significant vulnerability. Indeed, the decline in
agricultural activities has had several effects, including the disappearance of risk culture and the over-
urbanization of land. Real estate developers and investors have seized land to build, on marshes or dunes,
subdivisions, which are vulnerable to the risk of flooding (Duvat, 2011). Some of these lands ended up
under a metre of water when the storm passed, trapping the inhabitants in buildings unsuited to the
hazards. The second factor of vulnerability is the progressive replacement of populations of farmers and
sailors by urban dwellers, tourists and pensioners. These populations live from discontinuously on the
territory and therefore lose the knowledge of the natural functioning of the territory, leading to a
vulnerability of the populations. Xynthia was thus such a dramatic event because of "*the modes of
occupation of space (which) gradually neglected the hazards of submersion and flooding*" (Duvat,
2011). It is therefore no longer only natural disasters that impact cities but urbanization that leads to
over-vulnerability, leading to a melting pot of opportunities for risk amplification (Mitchell, 1999).
Concentration is thus perceived as an aggravating factor in risk management. This concentration is
expressed by the density of population present in a given territory. It is established that the denser the
area, the more vulnerable it is, the greater the potential for loss. It is therefore established that in urban
areas, natural hazards tend to have more serious consequences (Mitchell, 1999).
Three risks have a particular impact on urban areas (CRED and UNISDR, 2018).
• The earthquake is the most dreadful hazard, as it is responsible for the largest number of victims
worldwide, averaging 130,000 a year (Sigma, Swiss Re., 2011). However, the number of
victims depends very largely on the nature of the buildings and the nature of the preventive
measures. At the same magnitude, the disaster in Port-au-Prince claimed 222,000 victims, but
only 500 in Santiago de Chile. As for material damage, given the very unequal insurance
coverage, the official estimate is reduced to $10 billion in Haiti, but $30 billion in Chile. In
addition to its direct destructive effects, the earthquake can trigger either fires by breaking
energy networks, such as the one that ravaged Tokyo in 1923, or tsunamis.
• Flooding is also a major risk for large agglomerations that are located either in estuaries, on the
coast, in alluvial valleys or on slopes that have become unstable. Urban sprawl is often the most
vulnerable, due to poorly regulated urbanization, especially in areas where water is stagnant,
such as in Buenos Aires, Dhaka, Phnom Penh or New Orleans. It can also involve mudslides or
landslides on urbanized slopes such as the favelas of Rio de Janeiro. The urban dimension also
determines the extent of soil waterproofing, and therefore the extent of runoff. In addition to
these direct damages, there are also those related to the disorganization of services or the
degradation of equipment and industrial installations that are specific to any large urban area.
Climate change includes a new risk, that of the gradual rise in sea levels. As a consequence of
probable climate change, it threatens many of the world's port cities such as London, the Dutch
delta with Rotterdam/Amsterdam, but also Tokyo or New York.
• Wildfires, which can occur in periods of drought and heat waves, can cause immeasurable
damage, as in Australia in 2019, resulting in the destruction of 3500 homes, 5852 outbuildings,
34 direct deaths and 417 by excess from smoke inhalation (Borchers Arriagada et al., 2020). In
Europe, forest fires in Greece in 2007 and in Portugal 2017 claimed 80 and more than 100 lives,
respectively. In 2018, 99 lives were lost in Greece, 2,500 people were evacuated in Portugal and
Spain, 50 people evacuated in UK, while Sweden had to face the most serious series of forest
fires in its modern history, although with no fatalities.

The over-vulnerability of these urban areas in the face of natural risks also leads to the emergence
of "urban" risks.


### 2.2. Fragile urban spaces confronted with cascading effects

Urban space is made up of several infrastructures, some more essential than others. Called Critical Infrastructures (CI), these infrastructures concentrate all the functions (Pescaroli and Kelman, 2017) necessary for the proper functioning of a community. The term critical infrastructure only appeared in the United States in the 1990s following a succession of disasters, including the first attack on the World Trade Center (1993), followed by that of Oklahoma City (1995) and the gas attack in the Tokyo subway (1995). These infrastructures were then defined as vital to the point that "*their incapacity or destruction would significantly weaken (US) defense or economic security*". Critical infrastructure is defined as telecommunications, power generation systems, oil and gas storage and transportation systems, banking and finance, passenger transportation, water supply and distribution, emergency services (medical, police, fire), and those that ensure the continuity of government (Fekete et al., 2015). They are termed "critical" because their potential destruction could weaken the entire defense and economic organization (Serre and Heinzlef, 2018) of a country or city. Critical infrastructure can be natural; water supply, flood water storage; or physical; energy networks, telecommunication networks, emergency services, transport networks; or virtual systems such as cyber-information systems. However, these CIs interact with each other and thus create interdependencies (Serre, 2018) within the urban space. These interdependencies then play the role of a risk diffusion factor. According to the concept of the cascading effect (Bach et al., 2013; Nones and Pescaroli, 2016; Pescaroli and Nones, 2016; Serre and Heinzlef, 2018) i.e. a chain reaction causing changes in a territory some areas come to be impacted by the disaster, even if they were not located in the same area. directly in the flood hazard extension zone. As urban areas are interconnected, infrastructure failure will impact the territories across geographic and functional boundaries (Boin and McConnell, 2007). Because these components are connected at multiple scales, CIs can have an impact on much larger territory than their first impact territory. For example, floods can have an impact on a specific area, such as a road, but as the interconnected, the risk will spread to other territories that should not have been interconnected. naturally be flooded (Lhomme et al., 2013) by compromising power grids, supply of vital resources, etc. (Nones and Pescaroli, 2016). Therefore, some damages are not caused by direct physical damage, but by through business interruption. A distinction is made between direct and indirect impacts. The direct impacts are the tangible impacts and refer to the damage of the elements. physical (furniture, buildings, stocks, equipment, etc.). Indirect impacts occur when they are not caused by the disaster itself. Indirect impacts can be related to interruption or damage to critical infrastructure service. These may occur outside the area directly affected by the disaster and extend into the time after the shock (OECD, 2014).

### 2.3. The contribution of urban networks to the spread of risks

The role of urban networks is a good example for understanding and measuring what a CI failure can lead to. Urban networks are an essential part of the urban system. In an interconnected world, urban networks connect more and more people and territories and offer a wide variety of resources and opportunities. However, they also create complex situations of interdependence. Public transport, electricity, gas, telephone, heating, waste, etc. make the management of the urban system more complex. While they are essential for creating dynamics, relationships, and opportunities, they also create complex situations of interdependence. In addition to being a key component of the economy, these networks are also extremely vulnerable in the event of a crisis. Because of their interconnectivity, all urban operations depend on them. A single failure can have cascading effects affecting the entire network and, due to a reticular urban system, the whole city. Some examples illustrate these effects:

- Hurricane Katrina (2005) highlights the devastating effects of CI failure and related domino effects (Pescaroli and Kelman, 2017). The hurricane in August resulted in the breaching of protective dykes causing the destruction of 300,000 homes and 1,833 deaths (Knabb et al., 2005). The disaster was exacerbated by the domino effects that followed the destruction of the dikes, which made relief operations more complex. Transportation such as highways and


bridges were affected, reducing the ability to deliver vital resources - such as water, food and
medical supplies. Medical facilities were, for the most part, damaged or destroyed. All of these
effects have made the territory and its inhabitants more fragile, making it more difficult for CIs
to be brought back into service, but also for social and spatial functioning to function properly.
•   The aftermath of Hurricane Sandy in New York City in 2012 is a good example of these extreme
vulnerabilities aggravated by IC failures. Hurricane Sandy is one of the largest hurricanes ever
recorded in the Atlantic (Mitigation Assessment Team Report, 2013). New York University's
Langone Medical Center was evacuated after the generators failed due to flooding, causing the
transfer of 200 patients (Mitigation Assessment Team Report, 2013). The destruction of power
grids left 21.3 million people without electricity, and the blackout caused fires that destroyed
111 homes and damaged 20 others (Kunz et al., 2013). Daily life was severely disrupted by the
interruption of the metro, the breakdown of the heating network, security systems and
telecommunication services. In addition, alternative solutions such as emergency generators
could not be operated, as refineries were insufficient in number and unable to provide the
necessary fuel. While direct damage was estimated at 32.8 billion in repairs and restoration,
indirect losses cost the city and its citizens much more. The unpreparedness of managers and
citizens has considerably increased the impacts of the crisis. For example, the late evacuation
order and misinformation have resulted in the impossibility of evacuating certain institutions.
In addition, the crisis has put the vulnerability of sewer systems, poor anticipation of sewer
system failures, and the lack of network, the absence of a plan B for access to generators and
relay antennas, and the installation of the resistant flood barriers (Le Haut Comité Français pour
la Défense Civile, 2013) . In this case, the over-connected territory and society have created
new risks and made crisis and post-crisis management more difficult and complex

Societies and territories are therefore deeply vulnerable to potential functional disruptions due
to a crisis (Boin and McConnell, 2007). If the hazard persists (earthquake, flood, hurricane, etc.), it
is transformed by "*nature-society hybridization*" (Reghezza-Zitt et al., 2012), i.e. by the actions and
practices of humans in their environment. Thus, while natural hazards are not new, their impacts are
evolving due to climate change, urban growth and urban structural changes.
*2.4. The integration of the concept of multi-risk in the management of urban areas*
Due to the interconnection of territories and the emergence of cascading risks, risk management
must evolve from a single-risk to a multi-risk approach (Kappes et al., 2012) in order to understand the
diversity and consequences of interactions and interconnections. Whether due to a combination of
several natural risks, "*about 3.8 million km$^2$ and 790 million people in the world are relatively highly
exposed to at least two hazards, while about 0.5 million km$^2$ and 105 million people to three or more
hazards*" (Gallina et al., 2016) or as a result of cascading effects following a specific risk, or of man-
made disasters, territories and populations are exposed to a multitude of risks, forcing stakeholders to
innovate in traditional risk management. Furthermore, the climate change context increases the
likelihood of multi-risk exposure (Dilley et al., 2005; Komendantova et al., 2014). For instance, the
positioning of inter-tropical islands exposes them to a combination of risks such as storms, cyclones and
coastal erosion associated with the gradual rise of the oceans. If only one of these risks were analyzed
in a disconnected way from the others, the risk analysis, strategies and management established would
not be adequate and realistic to prepare these territories and their populations (Rosendahl Appelquist
and Balstrøm, 2014).

Risk management must therefore focus on integrated management in order to address the multitude
of interconnected risks. This comprehensive approach will allow considering their short- and long-term


impacts, which can have cascading effects, and to innovate in solutions adapted to an interconnected
world (Garcia-Aristizabal et al., 2012). Multi-risk assessments and all-hazards approaches need to be
strengthened, overcoming the limitation of single-hazards assessments in defining suitable and cost-
effective resilience measures in regions potentially affected by multiple sources of natural hazards. From
an operational perspective, multi-risk and multi-level (vertical/horizontal) governance frameworks
shifting from a single (siloed) risk focus to embracing a multi-risk approach when working with
technical and political authorities should be co-developed and co-evaluated.


The context of over-urbanization has led to a situation of vulnerability of urban spaces to risks. At
present, half of all people live in urban areas, a rate that is expected to reach 70% by 2050. This
concentration of people and goods weakens territories in the face of the growing increase in urban risks.
Because of the inter-connected world, the interdependence between the different urban systems (virtual
and/or physical), accentuates the dependence and vulnerability of populations and spatial functioning.
Some infrastructures, essential to the proper functioning of the territory, are more targeted. Faced with
the potential disruption of one of these critical infrastructures, a chain reaction can occur and have a
lasting impact on territories that cross administrative borders. The city needs to be analyzed as "*a system
of systems, with each of those systems (e.g. communications, water, sanitation, energy, healthcare,
welfare, law and order, education, businesses, social and neighborhood systems) potentially having
separate owners and stakeholders*" (UNISDR, 2017). The collaborative process underlying an
assessment of systemic vulnerabilities emerging from such an interpretation lays the foundations for
expanding the risk assessment framework towards wider objectives linked to the resilience of urban
systems in a multi-risk perspective (UN-Habitat, 2017). In the face of these growing uncertainties, risk
management must evolve and provide local managers and decision-makers with the keys to solutions.
New concepts are therefore gradually being integrated into risk management in order to help territories
and populations adapt to climate change, growing risks and related uncertainties.

## 491 3. Urban resilience: advances and limits

Faced with these growing challenges related to risks in urban areas, risk management has therefore
evolved by adapting the concept of resilience to the analysis of risks in urban environments.

### 494 *3.1. Urban resilience*

Urban resilience can therefore be defined as the concept that studies urban systems, i.e. the
interactions between the different components that participate in the creation of the territory. Urban
resilience refers to a systemic approach that encompasses the multiple layers (built, social, political,
etc.) and structures that produce an integrated vision of the urban object. Urban resilience would
therefore be a tool for analyzing the complexity of the urban system and defining the different capacities
and capabilities of each element that defines this system in order to live and survive a disruptive event.
The ability to define what is meant by resilience is an essential prerequisite for reducing the
consequences of a disaster. Determining what is "at risk" in a specific area is an essential step in this
regard. But when we talk about urban resilience, aren't all elements are essential? Most research on
operationalizing resilience focuses on a technical-functional approach (Table 1). As a result, it is mostly
the technical and material elements, such as urban networks, that are analyzed in these studies (Gonzva
and Barroca, 2017; Lhomme et al., 2013; Serre, 2018, 2016). However, an urban system is made up of
multiple components that are constantly interacting. There is no conceptual and theoretical consensus
in the scientific and policy community (Table 1) on the definition and objectives of urban resilience,
which reinforces the lack of clarity in establishing resilient risk management strategies.


| Sources | Systems | Definitions |
|---|---|---|
| OECD | Cities | Resilient cities are cities that have the ability to absorb, recover and prepare for future shocks (economic, environmental, social & institutional). Resilient cities promote sustainable development, well-being and inclusive growth |
| C40 | Cities | Cities are at the forefront of experiencing a host of climate impacts, including coastal and inland flooding, heat waves, droughts, and wildfires. As a result, there is a widespread need for municipal agencies to understand and mitigate climate risks to urban infrastructure and services – and the communities they serve. |
| ICLEI | Cities | A resilient city is prepared to absorb and recover from any shock or stress while maintaining its essential functions, structures and identity as well as adapting and thriving in the face of continual change. Building resilience requires identifying and assessing hazard risks, reducing vulnerability and exposure, and lastly, increasing resistance, adaptive capacity, and emergency preparedness. |
| Resilience Alliance | Cities | A resilient city is one that has developed capacities to help absorb future shocks and stresses to its social, economic, and technical systems and infrastructures so as to still be able to maintain essentially the same functions, structures, systems and identity. |
| Alberti et al., 2008 | Cities | The degree to which cities tolerate alteration before reorganization around a new set of structures and processes |
| Campanella, 2006 | Cities | The capacity of a city to rebound from destruction |
| Lamond and Proverbs, 2009 | Cities | Encompasses the idea that towns and cities should be able to recover quickly from major and minor disasters |
| Lhomme et al., 2013 | Cities | The ability of a city to absorb disturbance and recover its functions after disturbance |
| Urban Resilience Hub | Urban system | The measurable ability of any urban system, with its inhabitants, to maintain continuity through all shocks and stresses, while positively adapting and transforming toward sustainability |



| Holling, 1973 | System | The persistence of relationships within a system, a measure of the ability of systems to absorb changes of state variables, driving variables, and parameters, and still persist |
|---|---|---|
| UNISDR | System | The ability of a system, community or society exposed to hazards to resist, absorb, accommodate, adapt to, transform and recover from the effects of a hazard in a timely and efficient manner, including through the preservation and restoration of its essential basic structures and functions through risk management |
| 100RC | System | The capacity of individuals, communities, institutions, businesses, and systems within a city to survive, adapt, and grow no matter what kinds of chronic stresses and acute shocks they experience |
| Pickett et al., 2004 | System | The ability of a system to adjust in the face of changing conditions |
| Godschalk, 2003 | Critical infrastructure networks | A sustainable network of physical systems and human communities |
| Serre et al., 2013 | Critical infrastructure networks | Urban resilience aims to maintain urban functions during the event and recover thanks to resistance capacities (assessing damages), absorption capacities (assessing alternatives) and recovery capacity (assessing accessibility) |
| Cimellaro et al., 2010 | Critical Infrastructure | Resilience is defined as a function indicating the capability to sustain a level of functionality or performance fora given building, bridge, lifeline networks, or community, over a period defined as the control time that is usually decided by owners, or society |
| Ouyang et al., 2012 | Critical Infrastructure | Resilience as the joint ability of infrastructure systems to resist (prevent and withstand) any possible hazards, absorb the initial damage, and recover to normal operation |
| Longsttaff, 2005 | Community | The ability by an individual, group, or organization to continue its existence (or remain more or less stable) in the face of some sort of surprise |
| Adger, 2000 | Community | The ability of communities to withstand external shocks to their social infrastructure |
| Ganor, 2003 | Community | The ability of individuals and communities to deal with a state of continuous, long term stress; the ability to find unknown inner strengths and resources in order to cope effectively, the measure of adaptation and flexibility |




| Coles, 2004 | Community | A community's capacities, skills and knowledge that allow it to participate fully in recovery from disasters |
|---|---|---|
| Wagner and Breil, 2013 | Community | The general capacity and ability of a community to withstand stress, survive, adapt and bounce back from a crisis or disaster and rapidly move on |
| Asprone et al., 2014 | Hybrid approach | City resilience is based on the efficiency of hybrid networks composed by citizens and urban infrastructures. |
| Heinzlef et al., 2020 | Hybrid approach | The ability of populations, territories and infrastructures to put in place resources, skills and capacities in order to best experience a disruptive event so as to limit its negative impacts. Capacities can be both tangible (urban networks, supply of vital resources, etc.) and intangible (knowledge of risk, economic dynamics, institutional framework, etc.). |


*Table 1: Comparison between different system of study to analyze urban resilience*

*3.2. A complex urban system…*
One of the reasons for this lack of clarity is the complexity of current urban systems. The city
is a complex object to define, describe and analyze. The urban components are and the models are
struggling to analyze the urban system. Urban growth accompanied by urban, social, technical, political
and economic changes are leading to a fragmentation of urban space. This fragmentation and increasing
complexity does not and shared knowledge of urban space, which is a prerequisite for a global and
shared vision and knowledge of the urban space. complicates risk management.
Since the 1970s, systems thinking has emerged to address complex systems. The difficulty of
defining the city as an object emphasizes complexity and therefore suggests that we can consider the
city as a system. A system can be defined as a set of elements and interactions between elements that
form an organized whole, with the internal organization of the system constituting its structure and the
behavior of the interacting elements defining its dynamics. Each system is defined according to a
purpose and an objective.
This urban system (Bretagnolle et al., 2006) is defined by the interdependencies existing
between the various components of the city, due to the multiple networks of relationships they have with
each other. The systemic approach aims to observe, interpret and reconstruct the real world by putting
forward hypotheses on the organization of cities (Paulet, 2009). The analysis of a system is therefore a
construct and presupposes choices among the different variables of the system. The study of a city today
therefore implies understanding and considering the interdependencies between cities and their
components, and analyzing their connections.
The city thus first of all creates interweaving urban components, such as technical systems (such
as urban networks and/or critical infrastructures) or public infrastructures (governance, education,
health, police, justice, etc.) (Lhomme et al., 2013), but it also creates interrelationships with its
environment, due to its open system characteristic. As an open system, it both transforms itself through
intrinsic capacities but also receives resources through flows and information from their environment.
Since it is not self-sufficient, the relationships between cities and countryside, between cities and towns,
are essential and must be analyzed in the global study of an urban system. These interactions can
therefore be considered as a source of wealth, more (food) resources, knowledge, techniques, but also a
source of fragility (uncertainties, overproduction of waste, new urban risks, social risks, etc.). This non-
exhaustive list nonetheless allows us to understand the fragility of urban environments, because their
sources of growth, expansion and wealth can, at its peak, also be synonymous with vulnerability.
The city is therefore a complex object to apprehend and study. Because of its construction
protean, the city is difficult to define and identify as a single object. The evolutions are constant and



vary according to the social, urban and technical components, environmental, political, economic, etc., of the urban space. Focusing on the issues at stake challenged by risk, cities are also concentrating resources to deal with it. This is so in these spaces that urban systemic resilience must be analyzed and operationalized.

### 3.3. ... Including some limits

Despite its growing importance and use in expert and policy discourse, the concept of resilience faces many limitations.

#### 3.3.1.A conceptual vagueness

The concept of resilience faces a conceptual confrontation in the multitude of definitions and associated notions. This concept is today over-used, over-solicited in multiple fields (psychology, ecology, political science, physics, geography, etc.) and related to many concepts (Emrich and Tobin, 2018). This multitude of uses has turned it into a buzzword (Reghezza-Zitt et al., 2012), a word "*suitcase*" (Rufat, 2015) that complicates its understanding. A resilient system is in turn defined as a system capable of stability but also of adaptation and evolution (Hegger et al., 2016; Tempels and Hartmann, 2014). We speak of both "bouncing back" to a (potentially anterior) equilibrium or "bouncing forward" to a new state of balance and harmony. Faced with this ambiguity, or even contradiction, among the objectives and guidelines of resilience, actors and experts come up against grey areas (Disse et al., 2020). Beyond these two characteristics, Brand and Jax (2007) analyzed the studies, definitions and methodologies addressing the concept of resilience over the past 35 years and pointed to the abstract trend of resilience. According to them, resilience must therefore be perceived and understood as a perspective (of planning, risk management, spatial and social development) rather than as a concept or tool to be clearly and unanimously defined (Kim and Lim, 2016).

#### 3.3.2.A political reappropriation

This conceptual vagueness has contributed to the political reappropriation (Béné et al., 2018) of the concept of resilience without resulting in clear strategies adapted to local actors and territories at risk (Bahadur and Tanner, 2014; Béné et al., 2012; Cannon and Müller-Mahn, 2010; Duit et al., 2010) . Many scientists and experts have denounced the tendency to overuse and abuse the term resilience. Having become a political and management imperative, resilience has been transformed into a political and crowd-unifying tool. Resilience can therefore be used more for political positioning or institutions to strengthen their dominant governance model without necessarily leading to reflection on processes of transformation or evolution that are generally necessary for the establishment of resilient systems (Béné et al., 2018) .

Beyond limitations related to the lack of consensus on the concept of resilience, there are also limits to its implementation in risk management strategies.

#### 3.3.3.Financial limitations

The cost of a resilient approach or accommodations is often pointed out. Whether it is spatial redevelopment (reworking urban density, refuge areas, critical infrastructures, risk areas, etc.) or the purchase of so-called resilient development tools (Heinzlef et al., 2020), local managers and actors are faced with a mismatch between the cost of this approach and their daily priorities. The fact also that climate change and the associated risks are a more or less distant threat and hardly imaginable threat, makes decision-makers less focused on the necessary evolution of risk management strategies through the integration of resilience into the planning process (Leichenko et al., 2015).




### 3.3.4. Cultural barriers

The local cultural dimension of risk management can also be seen as a barrier (Heinzlef et al., 2020b) to the implementation of the concept of resilience (Heinzlef et al., 2019a). This socio-cultural dimension can be expressed at several levels.

At the level of local actors (Amundsen et al., 2010; Dilling et al., 2015; Kettle and Dow, 2014; Mozumder et al., 2011; Runhaar et al., 2012), it can be expressed through the culture of risk. The risk culture can be associated with the historicity of disasters on a specific territory and therefore by the succession of management strategies put in place to deal with them. Changing them can be complicated, especially if it requires new human and financial investments.

At the individual level, this cultural resistance or lack of understanding is regularly linked to a lack of awareness of the risks linked to climate change and a fear of changes in their habits and living environment (Amundsen et al., 2010; Measham et al., 2011).

### 3.3.5. Technical limitations

When resilience becomes an operational object, it often requires technical management tools. However, the general bias for operationalizing resilience involves its quantification and representation. Simply put, the tool must be able to conclude whether or not the territory is resilient. Numerous studies have provided answers to this issue. After establishing the need for urban technical networks in the functioning of urban territories, concluding that these networks contribute to the resilience of urban areas becomes obvious. A great deal of research has therefore analysed resilience through the resistance of urban networks (Barroca and Serre, 2013; Gonzva et al., 2017; Gonzva and Barroca, 2017; Lhomme et al., 2013; Serre, 2018). These approaches focus on the resilience of networks, critical infrastructures and the built environment, but very few address the concept in a global and systemic way. This technical-functional positioning leads to a narrow vision of the systemic spatial complexity. As a result, they only partially transcribe the spatial reality of urban dynamics and interconnections.

Faced with the difficult consensus around the concept of resilience, its operationalization is regularly questioned. The difficult formalization, linked to the multitude of interpretations and approaches, results in a complex transition from theory to practice. However, this is the challenge posed by all studies on resilience, in order to use this concept to build adequate risk management strategies. Several approaches have therefore attempted to respond to these challenges by proposing methodologies that aim to operationalize resilience. This operationalization translates into the design of tools for measuring resilience, spatial decision support systems or approaches that promote collaboration between experts and local stakeholders. In this section, we will analyze some of these works, frameworks, structures and methodologies. Some authors have attempted to synthesize all existing models (Constas et al., 2010; Schipper and Langston, 2015) but the forty or so models mentioned (Bahadur et al., 2015) underline the (over)abundance of approaches to resilience. We will attempt to scan the approaches aimed at assessing resilience through the creation of indicators, models proposing a conceptual framework or decision support systems, and then methodologies aimed at creating collaborative work in order to operationalize resilience.



## 4. Methods and tools for evaluation, modelling and integrating resilience into risk management

A large part of operationalization involves determining how a concept can be measured (Adger et al., 2004) and determining which indicators will be used to measure the concept in order to generate data about it. The assessment of resilience therefore essentially involves its measurement (Heinzlef et al., 2020a) and the creation of indicators.

### 4.1. Assessing urban resilience

#### 4.1.1. Taking advantages from indicator sets

Indicators are quantitative variables intended to represent a characteristic of a system or concept. They have been used to inform decision making, improve stakeholder participation, build consensus, explore underlying processes, etc. (Parris and Kates, 2003). The objective of an indicator is to provide information that should help actor to steer the course of action towards the achievement of an objective or to enable him to evaluate the result. The indicator can be a parameter, a value, a data or an observation. Its objective is to give indications or describe a phenomenon, a situation, an environment or a process. It is necessary to define a preliminary objective to which the indicators will tend. An indicator can be composed of a single variable or a combination of variables (Birkmann, 2006).

Regardless of the word used, the indicator primarily defines the compelling relationship between the information contained in the indicator and the object pointed to by the indicator (Birkmann, 2006). The function of the indicator is therefore to show, to place in space, and it is this spatializing nature that makes it interesting as a geographical tool (Freudenberg, 2003). If the indicator in the primary sense of the term does not analyze or define, it takes on its full meaning through the observer's reading of it. Because of its eminently operational nature, it enables observations and results to be anchored in practical reality. It answers directly to the question asked by the user confirming or not the initial hypothesis. The hypotheses and judgements made when choosing the questions and data relevant to the development of the indicator, as well as the evaluation of the usefulness of the indicator, require the existence of objectives, implicit or explicit. An indicator collects data and information in order to aggregate knowledge, which is essential for making the right choices (Wisner and Walter, 2005). For this reason, indicators are fully involved in the decision support process (Tate, 2012).

However, despite its operational nature, the indicator is only an experience of reality and not a proper experiment. It is therefore necessary to bear in mind that it is a practical image of reality but that it is not objectively the reality of the territory. The indicator merely reproduces or reconstructs an image of the geographical space, which makes the choice of indicator, variables and treatments very subtle and complex. However, this choice is itself built around representational a priori, a socio-cognitive paradigm that cannot be denied. It is therefore necessary to make the construction of these variables and indicators as objective as possible in order to claim that the results are real. There are several "formats" of indicators. Multiple indicators, for example, can be combined with unstructured composite indicators, or indices, which attempt to distil the complexity of an entire system into a single measure. Social indicators have been used since the 1960s, with applications to the environment (1970s), sustainability (1990s), and more recently vulnerability (Birkmann, 2006; King and Macgregor, 2000) and resilience (Cutter et al., 2010). The main global indices and recent regional studies that model various aspects of vulnerability include the vulnerability index, the Human Development (UNDP), the Disaster Risk Index (UNDP 2004) or the Disaster Risk Index (UNDP 2004). the Environmental Sustainability Index. The Social Vulnerability Index (Cutter et al., 2003) is the best known index for evaluation at the national level, with applications in the United States, Canada and the United States (de Oliveira Mendes, 2009; Finch et al., 2010; Holand et al., 2011).

#### 4.1.2. Resilience indicator challenges

Measuring resilience has become an international priority in order to build strategies for the future. risk management (Winderl, 2014). The question of how to measure resilience is as old and as important as the concept itself (Prior and Hagmann, 2014). Numerous indices and indicators of resilience have been developed in various disciplines. In general, they are used for different purposes and, as a





result, they measure different things. An exploration of attempts to measure resilience reveals the difficulty in establishing a measure that is both accurate and "*fit for purpose*" (Hinkel, 2011). Measurement requires that a phenomenon be observable and allow for systematic attribution of value, but the conceptual nature of resilience makes this difficult. Scientists do not have not yet agreed on specific conventions for measuring resilience and, consequently, there is a substantial literature that discusses both how and whether the phenomenon can and should be measured (Hinkel, 2011).

The identification of resilience requires planners to identify variables that trigger disturbances in a city (a community, region or landscape), the frequency and intensity of these events, and the mechanisms that enhance adaptability that can be activated to respond to (or avoid) these disorders. It is need to assess the socio-economic dimensions of an urban area (Ahern, 2011). As established previously, it is necessary to establish common denominators that induce vulnerability or strengthen resilience (Gonçalves, 2013) . However, the difficulty essential is to measure these dimensions. The significant challenges in measuring the resilience lead either to imperfect quantified measurements or to a search for indicators of universal resilience (Hallegatte and Engle, 2019). Cutter et al. (2008) highlight this difficulty in believing that "*if we conceptually or sometimes intuitively understand the vulnerability and resilience, the devil is always in the details, and in this case, the devil is measurement*" (Cutter et al., 2008b).

### 4.1.3. Examples of resilience indicators

- The Baseline Resilience Indicators for Communities (BRIC) (Cutter et al., 2014)

Cutter et al. developed BRIC (Baseline Resilience Indicators for Communities), which aims to define resilience indicators to map the level of resilience across the United States. Dividing resilience into six indicators - social, economic, community, institutional, infrastructural and environmental - Cutter proposes to measure resilience (Cutter et al., 2014). Each indicator is divided into sub-variables such as education, age, language proficiency, employment rate, immigration rate, access to food, disaster training, social stability, access to health, access to energy, etc. (Cutter et al., 2014). Each variable has a positive or negative effect on community resilience. Data acquisition was an important issue. More than 20 data sets were obtained from the U.S. federal government through online data portals. Four datasets were obtained from NGO websites, two through a contact with the American Red Cross, and one from an open access data portal at a major press briefing. One data source was the Dun and Bradstreet's Million Dollar Database and required a paid subscription to acquire the data (Cutter et al., 2014). Once the data acquisition was completed, a processing work, "cleaning" of the data was necessary. The chosen method of treatment was applied to the Min-Max. The Min-Max normalization assigns a value of 0 to the minimum value and from 1 to the maximum value. All other values are scaled to between zero and one by subtracting the minimum value and dividing by the range (minimum subtracted from the maximum). While this method makes it much easier to compare between a large number of variables, the disadvantage remains that the final score is not a measure absolute value of community resilience for a single location, but rather a relative value of community resilience for a single location. in which several locations can be compared. This is why the proposed work is done at the US level and not at a finer scale or for an single year, not being a comparative work over several years.

This approach is a key work in the process of operationalizing the concept of resilience. In addition to the definition of resilience criteria, it also makes it possible to locate more or less finely the territories on which to focus efforts to increase territorial and social resilience. Its systemic approach to the territory (considering the elements that make up the territory) is completely adapted to risk analysis.

- The DS3 Model (Spatial Decision Support System) (Serre, 2018)

The DS3 Model has defined three capabilities to assess the resilience of urban networks to flood risk. Resilience is defined here as the ability of a system to absorb a disturbance and subsequently recover its functions. Three capabilities are assumed to determine the degree of resilience of these networks:


o    The capacity to resist: this consists of determining the material damage following a risk.
     It is considered that the more a network is damaged, the more likely it is that there will
     be a slower and more complex return to service;

o    The absorption capacity: it illustrates the fragilities and strengths of the network
     allowing to build alternatives to it following a component failure;

o    Recovery capacity: this represents the time required to return to service of the network
     and its components.

These capabilities enable the resilience of a city's urban technical networks to be defined and measured. The methodology was tested to assess the resilience capacity of the Hamburg district, Am Sandtorkai/Dalmannkai. Each resilience capacity was analyzed according to the components of the neighborhood, at its scale and then according to the interactions with its environment. Using this technical resilience measurement tool, the case study analysis identified interdependencies and potential domino effects at the neighborhood level. The definition of these three capabilities made it possible to analyze resilience over a long period of time, before, during and after a disturbance. The systemic approach here is defined by the analysis of inter-network interactions and interconnections in order to assess cascading risks in urban environments.

- An hybrid approach (Heinzlef et al., 2020a)

This research made it possible to develop three indicators for defining and measuring resilience in order to gain a comprehensive and exhaustive understanding of the concept. These indicators analyze the urban, social and technical resilience of a city (Heinzlef et al., 2019a).

o    The social resilience indicator illustrates a population's ability to adapt and recover from
     disruption (Hutter and Lorenz, 2018). Many factors contribute to social resilience,
     including age (Cutter et al., 2010), community and political investment (Voss, 2008),
     socioeconomic status (Flanagan et al., 2011), knowledge and perception of risk, etc.
     (Hutter and Lorenz, 2018). This methodology understands social resilience as
     community resilience (Wilson, 2013) and not individual resilience (Hutter and Lorenz,
     2018).

o    The urban resilience indicator includes all urban dynamics, such as physical elements
     (Norris et al., 2008; Opach and Rød, 2013) (age of the building, urban density,
     building functions, critical infrastructure, etc.) or more virtual elements such as
     economic dynamics through the creation or suppression of businesses or touristic
     dynamism (Tierney, 2014).

o    The technical resilience indicator includes urban networks (Serre, 2016). It is used to
     analyze the diversity and accessibility of these networks within a radius of 100m in
     order to assess their resilience and their impact on the territory in the event of a crisis
     (Heinzlef et al., 2020a).

This study has been tested and validated in Avignon (France), and built with 90% open data in order to allow the reuse of this methodology on other national case studies. The systemic approach is illustrated by taking into account the multitude of elements that make up the urban territory in order to have a global vision and approach to the territory, its population and its potential resilience.

*4.2. Modelling resilience*

4.2.1. The usefulness of space-based decision-support systems

As the concept of resilience is a complex subject to address and operationalize for local actors, many tools have been created to simplify, define, measure and attempt to operationalize this concept. The need to create decision-support systems makes sense in terms of the abstraction of the concept. In risk management, taking is a complex combination of knowledge management and decision-making


processes. reasoning (Tacnet et al., 2014). Decision-support systems are defined as integrated computer
systems, designed for decision making. When territorial issues are addressed, these are referred to as
spatial decision Support System (DSS). They combine spatial and non-spatial data, functions analysis
and visualization of Geographic Information Systems (GIS) and decisions in order to construct, evaluate
and produce solutions (Keenan and Jankowski, 2019). These space-based decision support systems have
been developed to address the limitations of the GIS such as lack of modeling capabilities and lack of
flexibility of GIS for adapt to variations in the context or spatial decision-making process (Densham,
822 1991).

4.2.2.The integration of geo-visualization techniques
Indeed, current tools such as GIS are often inadequate in the face of the complexity of the real
issues facing users (Andrienko et al., 2007). For individuals, the visual context favors the acquisition of
knowledge (Kwan and Lee, 2003). There are many forms of data visualization that are primarily
scientific and information visualization (Marzouki et al., 2017). If data have a combination of spatial,
semantic and temporal dimensions, they are referred to as geographic data/information and geo-spatial
data (Marzouki et al., 2017). The visualization of these data then becomes specific, and goes beyond
simple scientific and information visualization (Kurwakumire et al., 2019). The integration of
visualization in the analysis of geo-spatial data has led to a transformation of traditional mapping
through the digital era (Çöltekin et al., 2017) . This evolution of traditional mapping has led to
geovisualization, a "*set of visualization methods and tools for interactively exploring, analyzing and*
*synthesizing location-based data for knowledge building*" (Dykes and International Cartographic
Association, 2007). Geovisualization combines scientific visualization, information visualization,
mapping, geographic information systems (GIS), exploratory data analysis and many other methods to
explore, analyze, synthesize and represent geographic data and information (Nöllenburg, 2007) . As a
result, many spatial decision support systems have been equipped with visualization techniques and
dynamic interfaces to combine technological capabilities with local interpretations and knowledge. Map
production is accessible and understandable through a visual interface to enable exploration,
understanding, analysis and reuse of a complex, geolocalized and heterogeneous database.
Thanks to a dynamic interface and a technical power capable of processing complex data,
geovisualization tools allow to communicate information about complex data necessary for the decision
support process. In addition, the interactivity of these tools allows the users to be actors in front of the
tool, by navigating, by making a visualization request, by downloading data or displaying information
as needed. The tools of are therefore both communication tools and tools for the production of
geovisualisation knowledge by being an integral part of the "*reflection/knowledge process*".
(MacEachren et al., 2004). As Bishop et al.(2013) point out, neuroscience has been a major contributor
to the development of the human brain. and has already demonstrated that visualization techniques are
essential to cognitive processes. leading to decision making (Padilla et al., 2018). Geovisualization thus
integral part of spatial decision support systems, as it allows to meet both scientific and societal needs
to initiate a process of reflection and thereby build and produce knowledge.
Several methodologies have produced tools to clarify the concepts of resilience and
vulnerability. These tools are spatial decision support systems and have made it possible to dissect the
concept of resilience. The objective of each of these approaches is to make the concept accessible by
creating links between scientific advances and territorial reality.
4.2.3.Examples of spatial decision support systems
• The DOMINO tool (Robert et al., 2008)
A tool for modelling the spatial and temporal propagation of domino effects between critical
infrastructures (CI) has been developed for the city of Montreal. It consists of a geographic database in
which organizations have entered relevant information about their dependencies on the critical resources
they use. Modules, created on the structure of the expert systems, combine information from different
organizations in order to identify interdependencies between them. A time simulator has also been


developed to visualize the propagation of potential domino effects following a failure. Being a geomatics
tool, it is It is possible to combine several layers of information in DOMINO. The partners' managers
(CI managers and emergency preparedness officials) have secure access to it, managed according to
levels corresponding to their user profile. Thus, each organization has access to its information, while
the results of the simulations are available to all the users. In terms of resilience, DOMINO allows
analyses on the different parameters of the resilience. Its establishment on a territory requires the various
organizations concerned to exchange information on their own disruption management capability. The
implementation of this information enables consistency analyses to be carried out on a given territory
and integrate broader community implications. The systemic dimension of this tool lies in the fact that
it analyses the interdependencies of critical infrastructures in an urban space and models potential
service disruptions.
• The ViewExposed tool (Opach and Rød, 2013)
A Norwegian study addressed the issue of vulnerability of territories s response to climate change
(Opach and Rød, 2013). In order to avoid an increase of local and national vulnerability, the researchers
have developed a ViewExposed tool, including the aim is to inform local authorities about the most
vulnerable areas of the territory Norway and also the causes of this vulnerability. The methodology used
is based on the work of SoVi (Cutter et al., 2003) and the University of South Carolina (Tate, 2012).
Several steps were necessary to create this tool:
o Creation of vulnerability indices for storms (StoVI), floods (FloVI) and landslides
891         (SliVi)
o Work on data and indices to create a compiled physical vulnerability index (PhyVI)
o Assessing Norway's social vulnerability and creating a Social Vulnerability Index
894         (SoVI)
o Compilation of the Physical and Social Vulnerability Indicator to create an Integrated
896         Vulnerability Index (IntVi)

The objective of IntVI is to focus on a municipality's exposure to natural hazards and to put it into
perspective with regard to the local population's capacity to resist them. For PhyVI, the exposure of
municipalities to natural risks is expressed as a percentage and depends on the work of Norwegian
insurers (Norwegian Natural Perils Pool). Based on the data, the researchers were able to determine that
during the period 1980-2010, 60% of the damage was caused by storms, 26% by floods, 7% by
landslides and 5% by storm surges (Opach and Rød, 2013). Concerning SoVI, the objective was to
assess the adaptive capacity of municipalities with regard to physical exposure. Thus, in the next step,
the SoVI was calculated using the methodological framework constructed for Norway by (Holand et al.,
2011). Finally, PhyVI and SoVI were compiled to create IntVI. To do so, the weights most correlated
with the Norwegian Natural Perils Pool claims data were used: 60% for PhyVI and 40% for SoVI. The
tool, which takes the form of an interface, has been created for professionals, local elected officials and
residents. It is the product of a collaboration between scientists and local experts through workshops
(Opach and Rød, 2013). The authors of the interface wished to answer two fundamental questions: Who
are the vulnerable people? And where do they live? The objective is therefore to identify regions with a
high level of social vulnerability to environmental risks in order to reduce it and thus help to improve
their resilience. In addition, the platform is open and scalable as any actor in the field can submit a
reflection using the "submit a comment" section of the interface. Although focused on the concept of
vulnerability, this tool also integrates the response of local managers and actors to natural disasters. It is
therefore both the vulnerabilities but also the resilience strategies that are integrated. In addition, this
tool proposes a collaborative and participatory approach between local actors and scientific experts.
*4.3. Integrating resilience into urban management through collaborative approaches*
The United Nations International Strategy for Disaster Risk Reduction (UNISDR) has developed
10 key points for creating resilient cities. The first point is to set up organizations or coordinations to
understand and reduce risks, based on the participation of citizens and civil societies (UNISDR, 2015).





The objective is to build local actions and alliances to ensure that each actor understands his or her role
in reducing and preparing risk reduction and resilience strategies (Heinzlef et al., 2020b; Gupta et al.,
927 2010).

4.3.1.Collaborative approaches, a key for operationalizing resilience
Involving "local" people or people directly concerned by the issues studied does not appear to be
new (Toubin et al., 2015) and even less original. The richness of having people from all walks of life
interact with each other facilitates an exploration of possibilities, enriching discussions, encouraging
cross-fertilization of views on the same subject, making it possible to be both more measured and more
incisive in a specific area. The contribution of "profane" knowledge in thorny social and societal issues,
as scientific knowledge cannot respond to all uncertainties, with the result that "expert" conclusions are
called into question. Resilience, a social and thorny concept, is therefore a subject that would require
the confrontation of views, knowledge, scientific and practical knowledge, perceptions and
interpretations. However, although the population is often the first to be impacted by natural hazards
and their inappropriate management, the fact remains that the inhabitants (Kuhlicke et al., 2011) and
also the urban services (Toubin et al., 2015), which are nonetheless first-rate actors, are not sufficiently
involved. The defended idea is that the creation of a hybrid knowledge (Djenontin and Meadow, 2018;
Lemos and Morehouse, 2005; Schneider and Rist, 2014) allowing the involvement of all actors of the
territory, from the inhabitant to the manager via the scientist, would make it possible to operationalize
urban resilience thanks to an appropriation of the concept and stakes of urban risks. In fact, collaboration
is mainly based on the appropriation of the different stakeholders of the same subject of tension and
discussion. Collaboration therefore goes beyond the simple exchange of knowledge and information,
but makes it possible to "*create a shared vision and articulated strategies for the emergence of common*
*interests that extend beyond the limitations of each particular project*" (Chrislip, 2002).
4.3.2.Examples of collaborative approaches
• Improving Urban Resilience through Collaborative Diagnosis (Toubin et al., 2015)
In her thesis, Marie Toubin develops a methodology to contribute to the improvement of conditions
of urban resilience and more particularly the resilience of urban networks. Her analysis of the
interactions and interdependencies of urban networks highlighted the intrinsic fragilities of urban
systems and their management in the event of a crisis. Faced with the challenges observed, the research
objective was therefore to develop methodological approaches and tools to help urban service managers
identify and characterize technical and organizational interdependencies in order to ensure service
continuity despite a disruption. The approach was built by integrating the main managers of the City of
Paris' urban services. The methodology made it possible to construct interviews to assess the criticality
of the resources required for the system to function properly. It was therefore possible to rank the
resources according to their importance and use. This research made it possible to draw up and analyze
the interdependencies of the Parisian urban networks. It highlights certain dependencies, particularly
those on electricity, telecommunications and travel. The collaborative approach made it possible to
involve managers in thinking about strategies to mitigate or at least manage these interdependencies.
Moreover, the collaborative process has illustrated the need to move beyond isolated approaches but
instead to foster a common vision. The interweaving of scales but also of services makes cooperation
and transparency between operators and decision-makers indispensable for the construction of a more
resilient city.
• Resilience by design in Mexico City: A participatory human-hydrologic systems approach
973       (Freeman et al., 2020)

The study developed by Freeman et al. (2020) in Mexico City highlights issues of building a
freshwater resilience of urban systems among several territories and stakeholders. In order to find a way
to manage systems of feedbacks and tradeoffs between stakeholders, Freeman et al. have developed a
Resilience by Design methodology (Brown et al., 2020). The aim of this methodology is to identify with



local stakeholders, resilience of what, to what; for whom and what can be done? Face to a complex
issue, this methodology provides a planning and common framework to identify solutions and
compromises between urban managers, political stakeholders and decision-makers. In this case study,
Resilience by Design methodology revealed "*consistent stakeholder preferences for social (such as*
*equity in water allocation among users) and economic performance*", such as domestic, agricultural and
industrial sectors. These common solutions guide to persistence, adaptation and transformations.
Understandings and choices about how much "resilience of what, to what, for whom and at what cost"
require a shared narrow and adaptive approach (Freeman et al., 2020). Thinking jointly about issues and
related solutions helps to establish an understanding of the concept of resilience and established
strategies over time. Actors must therefore debate and envisage solutions in an egalitarian and united
manner in an evolving territory in order to tend to increase its resilience.
Several methodologies exist in order to operationalize resilience concepts and integrate it into urban
risks strategies. The main approaches are divided into three categories: (1) assessing resilience through
its measure with indicators, (2) modeling resilience with geovisualization techniques and (3) developing
collaborative approaches in order to lead to resilience understanding and adoption by stakeholders.
Indicators are helpful to define main resilience characteristics and to provide a measurement to
analyze resilience potentialities. These indicators might be specific (Serre, 2018) or exhaustive (Heinzlef
et al., 2020a). They have an important utility to urban managers to define low resilience areas and
concentrate their strategies on it.
Geovisualization techniques are used to unbuilt resilience abstraction thanks to tools, interfaces and
data which allow comprehension and facilitate resilience integration. Interactivity, communication,
navigation, visualization lead to a precise resilience analyze. These tools are essential for knowledge
construction and sharing and are part of the "reflection/knowledge process".
Finally, collaborative approaches lead to local stakeholders' responsibilities to integrate resilience
into risk strategies management. It is useful to create a shared vision on complex concepts and strategies
between "experts" and "local actors". Their proper experiences (local risk management heritage and
scientific knowledge) lead to a territorialized risk and resilience strategies. It is also a long-term
guarantee to resilience strategies adoption.

## 5. Discussion

The multitude of existing models for operationalizing resilience indicates the growing importance
of the concept. These models, as diverse and varied as they may be, are essential to the transcription of
the concept into a concept tool (Gonzva and Barroca, 2017). Going beyond the controversy over the
exact definition of the concept, these models propose to operationalize resilience. The accuracy of their
methodology then takes a back seat because what matters then is not that the model be rigorous, but that
it be operational. However, not everyone has the same objective or goal (Table 2).

| Names | Category | Approach | Systemic Approach | Intended Audience | Effective Appropriation |
|---|---|---|---|---|---|
| BRIC (1) | Indicators | Global approach | Yes | Decision-makers and urban managers | ++ |
| DS3 Model (2) | Indicators | Technico-functional approach | Yes | Critical infrastructure managers and urban managers | + |
| Hybrid Approach | Indicators | Global approach | Yes | Decision makers, urban | +++ |

| | | | | | |
|---|---|---|---|---|---|
| (Avignon case study) (3) | | | | managers and citizens | |
| DOMINO (4) | Spatial decision support system | Technico-functional approach | Yes | Critical infrastructure managers | ++ |
| ViewExposed (5) | Spatial decision support system | Global approach | No | Decision makers, urban managers, insurances and citizens | ++ |
| Toubin et al., 2015 (Paris case study) (6) | Collaborative approach | Global approach | Yes | Critical infrastructure managers and urban managers | +++ |
| Freeman et al., 2020 (Mexico case study) (7) | Collaborative approach | Global approach | Yes | Critical infrastructure managers and urban managers | + |


| | | |
|---|---|---|
| (1) | Cutter et al., 2014 | |
| (2) | Serre, 2018 | |
| (3) | Heinzlef et al., 2020 | |
| (4) | Robert et al., 2008 | |

| | | |
|---|---|---|
| (5) | Opach and Rod, 2013 |
| (6) | Toubin et al., 2015 |
| (7) | Freeman et al., 2020 |

*Table 2: Models' categories*

The diversity of these models illustrates the interest and efforts developed to respond to the challenges of operationalizing the concept of resilience. While some apprehend urban resilience through the analysis of networks and through a technical-functional approach, others seek to develop hybrid, more exhaustive approaches that attempt to understand and analyze the diversity of the urban territory. The decision support approach also differs from one tool to another, with some advocating the usefulness of indicators, others justifying the need for visualization to lead to a process of understanding and decision making, and finally, some defending the need to integrate local actors at the beginning of any reflection on the concept of resilience. These models are neither exhaustive nor exclusive and it is necessary to use them jointly or at different times and phases in the construction of a resilience strategy. However, this multitude does not promote the understanding and appropriation of a concept that is still abstract for many local actors and managers. Whether it is due to the overly technical nature of the tools (such as for the DS3 Model), a lack of understanding of the concept or even a lack of knowledge of the tools themselves, local stakeholders have very little appropriation of the operationalization methodologies and therefore the concept of resilience.

A tool to define, measure, clarify and assist in decision-making would therefore be of significant interest. The objective of a new tool can be used as a basis for reflection and suggestions for further progressive implementation of the concept of resilience in risk management strategies. This prototype would be to promote an inclusive approach that would make it possible to bring together the different existing approaches around the concept of resilience and to develop a framework for reflection and action between local actors and scientific experts around the issue of operationalizing the concept. This type of tool could be achieved through the design of a resilience observatory. Observatories are key tools to support the observation, reflection, understanding and analysis of phenomena or territories. These tools, which are at the interface of reality and knowledge, are essential in the decision-making process, allowing the acquisition of knowledge and data while taking the necessary distance to have the most global vision possible of a phenomenon. Their usefulness in establishing monitoring of phenomena, territorial evolution and interaction, make them essential tools for apprehending events over




the long term, which is essential for establishing resilience strategies. A team based at the Oceanic Island Ecosystems joint research unit (UMR EIO) of the University of French Polynesia (Heinzlef et al., 2020, 2019b; Serre et al., 2019) has launched a prototype observatory on the islands of Tahiti and Moorea to analyze, measure and operationalize resilience. The objectives are multiple (Fig.1) and focus in particular on increasing knowledge of territorial risks, the acquisition, storage and enhancement of data related to risks and resilience and finally the integration of stakeholders in the process of reflection and implementation of resilience strategies.

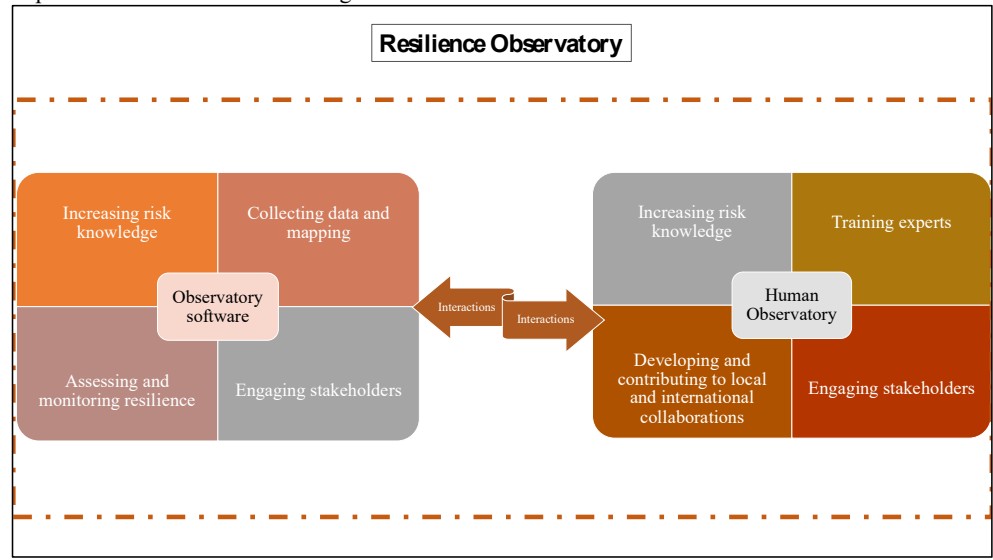

*Figure 1: A Resilience Observatory Prototype*

This prototype can serve as a basis for reflection and suggestions for further progressive implementation of the concept of resilience in risk management strategies.

## 6.  Conclusion

This article has provided a review of the concept of resilience and its operationalization. Confronted with a conceptual vagueness and a multiplication of definitions, notions and associated concepts, resilience loses its relevance and usefulness in risk management strategies. Yet this concept, which encourages adaptability, evolution and flexibility, is perfectly in line with climate change and the associated risks and uncertainties.

The currently challenge, whether in the scientific community or in urban planners and decision-makers sphere, is to work on its operationalization by promoting concept understanding and its adoption by local actors. This need has led to a multitude of scientific positions, tools and methodologies aimed at dissecting the concept of resilience and the concepts and capacities associated with it. These operationalization strategies can promote the design of indicators to define and measure resilience, develop spatial decision support systems to visualize territorial resilience or promote the implementation of collaborative approaches to involve local stakeholders in the integration of the concept in local risk management strategies. Although these methodologies in themselves provide opportunities for reflection or even initiatives for resilience strategies, their contribution remains modest and visible in a very short period of time.

Thinking about a new kind of tool for addressing resilience in the long term and an inclusive approach to the concept and associated methodologies would make it possible to respond to these current limitations. This tool, which would take the form of a resilience observatory, would make it possible to



develop a toolbox, bringing together conceptual and tangible advances related to the operationalization of resilience.

## 7. Acknowledgments:

This publication has received financial support from the CNRS through the MITI interdisciplinary programs and from the IRD.

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
