# Peer review of "Resilience issues and challenges into built environments: a review"

_Natural Hazards and Earth System Sciences, 2020_

## Referee Comment (RC1) · Anonymous Referee #1 · 30 Jul 2020

General Comments: The manuscript is a review manuscript and presents existing strategies and tools dealing with operationalization of the resilience concept in build environment. Manuscirpt detailed investigates the "resilience" concept from historical background till today with substantial amount of references. However some important improvements are required. Transitions from any level of titles to its subtitles are very hard. Many times section or sentence start without any prepatory information. In other words sentences comes out of the blue sky. Restructuring the whole manuscript is necessary. The manusript written mostly with "daily spoken English" with numerous typing error. Besides, due to the complexity of the subject sometimes you lost your way and very hard to follow. For a lighter text a native English check is strongly recommended.

Specific Comment: For historical development of the usage of "resilience" see Alexan-

der DE (2013). Resilience and disaster risk reduction: an etymological journey. Nat. Hazards Earth Syst. Sci., 13:2707-2716. There are some more details in using first or very early times of "resilience" term in scientific community. Introduction section very similar with Alexandr DE (2013). My suggestion: Re-organise in the way of shortening this section due to large overlap with the Alexander DE (2013) wich published in the same journal.

Technical corrections. Line 37-38: Rephrase as: "Despite its growing success, the operational relevance of the concept is therefore constantly being questioned." Line 42: "Over the past 20 years or so,..." Rephrase as "During last two decades" Line 62: The sentence too equivoke in it is meaning. "Interdisciplinary" itself, or "interdisciplinary approach/ interdisciplinary studies" can serve it. Make clear please. Line 245: Rephrase as "Faced with increasing risks, stakeholders have identified two concepts (Saunders and Becker, 2015); resilience (taking into account the management of disturbances) and sustainable development (analyzing the balanced economic, social and environmental development of the territory)." Line 307: change "metres" with "meters". Line 415-416: Remove the sentence. You already mentioned in the previous sentence. Another option: you can merge two sentences. Line 422: Replace "IC" with "CI" Line 428: Replace "metro" with "subway" Line 513: Replace subtitle of " A complex urban system..." with "Complexity of an urban system" Line 515: A short introductory sentence required. "...of this lack of clarity...." "this" refer what? Line 520: "complicates", use capital letter when you start to a new sentence.

Line 552: Replace subtitle of " ... Including some limits" with "Some important issues of limitations of resilience concept" Line 554-555: Remove the whole sentence Line 560: Replace "in multiple fields" with "in many fields". Line 632: This paragraph not connected with its subtitle. It seems to be belong the next section. If so, move there. Line 694: Remove "the Disaster Risk Index (UNDP 2004)." Connect "or" with "the Environmental Sustainability Index." Line 696: Two times you refer United States. Remove one of them. Line 702: Remove "." after "future" Line 727: Cutter et al. (2014) is better.

Line 728-730: Gramatically incorrect sentence. Rearrange! Line 746-47: Rewrite the whole sentence. A very difficult sentence to understand. Line 817-819 Many "and" in one sentence. Rearrange! Line 851: Remove "." Line 852: Remove "." Line 866: Critical Infrastructures (CI) already abbreviated in previous section. Use only the short form. Line 871: "it is" used two times. Line 884: "s"? Line 887 "territory of Norway" is better. Line 927: An Introductory sentence required for why you are going to set a series of subtitles. Line 953-988: You should start with introductory sentences. Then harmonize examples and/or approaches by citing references to present instead of give the summary of the study. Line 977: You already cite to Freeman et al (2020). Is design methodology adopted from Brown et al?. If so, rephrase. If not, clarify it. Line 991: No connection of the paragraph with the previous one and with its title. Line 992: Some references required. ". ...risk strategies (reference/s) Line 1005: In Discussion section it is expected from Authors to summarize basic arguments/findings in first paragraph after an introductory paragraph and continue with one or two more paragraph wich discuss "strong points" of the presented approach and "constrains". In this study you summarize and presented existing model in Table and give a figure about resilence observatory. I recommend to write this section under different title. In current form this is not "Discussion" of the presented study.

---

## Referee Comment (RC2) · Anonymous Referee #2 · 6 Aug 2020

**1   General comments**

This paper aims at providing a review on the concept of resilience specifically on the built environment.  Since the concept of resilience suffers from less acceptance in practice mainly because of its theoretical character and numerous definitions (as it is concluded), it is worth to give an overview on the various approaches.

However, from my point of view, the authors miss to bring clarity to the jungle of terms and concepts, which would make a significant contribution to the discussion on the concept of resilience. The paper is very long (too long from my point of view) and could be substantially reduced without missing the main goal. Some paragraphs describe general concepts, which are not directly related to the concept of resilience.  This carries

the risk that the reader looses the main track to the major issues of the paper.

The language of the paper has to be considerably improved to meet the quality standards of a scientific paper in a WoS journal. Beside numerous typing errors, paragraphs are often not linked to each other and the red thread gets lost.

The paper deals with a social science concept and although some links to natural hazards are given, the question arises as to whether NHESS is the right target journal for this paper. I would have expected more information on how the concept could help to improve risk management of natural hazards. At the very end, the authors argue that a tool would be needed but it remains vague, what such a tool should contain and how such a tool should look like. May be that for a theoretical review of this kind of concepts other journals would be more appropriate.

If the authors decide to publish this article in this journal, I recommend a major revision of the paper including restructuring and reducing the content, a reduction of the content and a closer focus on the main issues underlined with some examples.

**2  Specific comments**

**2.1  Introduction**

- line 37: What is the growing success? In the next line you write the concept is questioned. Isn't this a contradiction?

- line 49: at the end of the sentence, I think a reference would be needed.

- line 132: why "attempted resilience definitions"? Consider to change the section title.

- line 134: . . . many different disciplines . . . redundant to paragraph above.

- line 136: you probably mean the criteria for determining when a system has recovered.

- lines 147 – 157: check for redundancies and language. Which capacities did Serre (2018) mean?

- lines 218 – 219 very unspecific. What is positive?

- lines 236 – 237: I do not understand what you want to say here.

- lines 277 – 278: "The concept . . . " this has been said several times.

2.2 Urban risks: . . .

- lines 291 – 292 what exactly has increased? The number?

- lines 303 – 309: where do these numbers come from? (UN, 2018)?

- line 322: isn't it primarily the exposure that increased rather than the vulnerability? Of course both could have increased, depends on the definition.

- lines 328 – 330: . . . live discontinuously from . . . natural functioning . . . natural functioning: very general

- lines 335 – 337: repetitions, consider rephrasing; also very general.

- lines 339 – 364: is the description here really necessary? Same holds for lines 370 – 400; could be at least reduced.

- lines 401 – 489: the question would be here, how the concept of resilience could help to better deal with these complex systems. You mainly describe the complexity. I suggest to cut these sections to closer link it to resilience.

2.3 Urban resilience: . . .

- line 499 – 500: is urban resilience really a tool? Shouldn't a resilience tool be rather a tool for analyzing the complexity and ways to improve it. In this context (and for the whole paper): what the difference between resilience and robustness? I think this term should be mentioned and defined somewhere.

- table 1: citations should be consequently added in the left column.

- line 513 and 552: check the section titles and rephrase

- line 518 – 519: check the sentence

- line 539: difference between city and town?

- lines 559 – 561: this sentence better fits at the beginning and is repeated several times

2.4 Methods and tools . . .

I think this part is the most essential one, while the upper sections could be shortened as much as possible. I recommend to reduce general descriptions such the one on "indicators" (lines 669 –680) as much as possible.

- lines 813: Check sentence, same next line: "taking" is a complex . . . not clear what you mean.

- The section 4.2.2 is very generally and could be shortened to those aspects really relevant here.

- line 871: . . . it is It is . . .

- line 884: . . . territories s response

- line 886: . . . including . . . please check.

**2.5   Discussion**

I think this is not really a discussion. I would expect that you discuss which of the concepts are used in practice and when not (what I assume for most of them), what's the reason for that. Is it the concept itself or the way it is implemented? What is missing that the concept could be used and implemented in practice? Wouldn't figure 1 better fit in one of the upper chapters?

**2.6   Conclusion**

- multiplication → multiplicity

- . . . perfectly in line . . . you perhaps mean . . . is able to be used in the context of climate change and . . . Next line: currently challenge → current challenge

- The last sentence sounds very hypothetical and you could say more in the discussion how this tool could look like.

**3   Technical corrections**

- line 13/14: . . . innovate existing ? risk management strategies

- line 34: . . . such as physics . . .

- line 54: resiliences

- line 123: word identity double

- line 142: We evaluate . . .

- first reference: who is the author? Reference is uncomplete.

---

## Referee Comment (RC3) · Anonymous Referee #3 · 8 Aug 2020

The authors present a manuscript with a review on "resilience" of the built environment, focusing on an urban context. The current version of the manuscript shows some major weaknesses which should be removed before the work may become acceptable for publication in NHESS.

1 Introduction

The introductory paragraph (1) should focus more clearly on the overall paper aim (resilience and related challenges for the built environment) since operationalising resilience is a challenging issue with different disciplinary roots. As such, we could consider resilience as, for example, also "vulnerability", from a physical, social, economic or ecologic perspective (see e.g. the distinctions made in a recent textbook edited by Fuchs and Thaler, 2018). As such, the statement made in the introductory sentences

(...many disciplines, physics, psychology, ecology or risk management) remains arbitrary and lacks a solid distinction between disciplinary approaches ot be opposed (or discussed in the subsequent sections). Moreover, "risk management" is not a discipline but a method used by multiple disciplines! Moreover, the authors further argue that this "disciplinary and conceptual vagueness makes the use f resilience and its integration into risks" – which is neither grammatically clear, nor from a subject point of view (in particular because this "vagueness" has not been introduced before). Further, it is not clear why it is challenging to "move from theory to practice".

**1.1 Resilience at the crossroads...**

Again, this section starts with some strong statements which I cannot follow. While vulnerability is an integral part of risk management (e.g., International Standards Organisation, 2009), resilience is only if defined as the counterpart of vulnerability. As such, the introduction to section 1.1 needs careful revision, also with respect to the overall disciplinary use of the term resilience and related conceptualization in risk management. The same is valid for the subsequent sentence stating that the concept of resilience is "over-used", see the already cited book section of Emrich and Tobin (2018) – by the way, citation in the reference list is incomplete.

In my opinion the overall introduction to section 1 should also explain why the subsequent sections are focusing solely on physics, psychology, ecology and "risk management", and not e.g. also on social sciences other than psychology or economics. Moreover, the question is if we could use the ecological concept of resilience to explain observations in natural hazard risk management, such as e.g. the idea of "building back better" (see e.g. discussion in Papathoma-Köhle et al. 2019).

**1.2 Attempted definitions**

Here it would be nice to see a kind of table to better shown contradicting and similar characteristics of conceptualizing resilience, and to better show the understanding of the authors of different terms such as adapting of reacting to a threat – to my understanding these differences do not make a difference in defining resilience rather than explaining the capacities of affected systems at different stages of the overall risk management cycle. Would be nice to see some explanations here.

1.3 Concepts for perception As this manuscript should provide an overview on different pathways to resilience I am wondering why only one perspective (the one of social sciences) is taken in section 1.3.1 – it is clear that different disciplinary foci exist, but a review should provide an overview on the main concepts and as such I am missing at least a "physical" and "economic" approach here, and these can co-exist together with the socioscientific approach of seeing resilience and vulnerability not as counterparts but as additives in risk management (as vaguely stated in lines 235 ff.). This should be more elaborated.

A similar string could be followed in section 1.3.2

2 Urban risks. . .

As far as I understood the overall manuscript is centered on urban areas, as such this should be better reflected in the title (instead of "built environments"). Moreover, at least from section 2 onwards we would need a proper definition of how risk and vulnerability are understood by the authors so that the overall aim of providing a review on the ruse of different types of resilience can be better understood with respect to and opposed to the term "resilience" and the specific use with respect to urban areas (?). Sections 2 and 3 are then a bit abruptly focusing on critical infrastructure in urban areas, if this is the overall aim here also CI and even networks as part of CI should be mentioned earlier and mirrored in the title accordingly. Otherwise, the sections and paragraphs need to be better connected so that potential readers will be guided through the use of the term "resilience" in urban areas and with respect to critical infrastructure in cities or even networks.

2.4 Multi-risk

It remains unclear why the discussion on integrating multi-(hazard and) risk in the management of urban areas is necessary for the review on the resilience term. Needs clarification, also with respect to the challenge if we would like to assess multiple dimensions of resilience for multiple hazards or even cascading events. In line 462 the authors further state that "risk management must therefore focus on integrated management in order to address the multitude of interconnected risks", which is not the case if risks to be assessed are clearly defined (as such we can e.g. compute risk for flood hazards in an urban area only, neglecting any interconnections with other hazard types, but the overall result will not be wrong with respect to flood risk management, it will only be incomplete because not all hazards that may occur have been assessed. Nevertheless, this quite often the case in daily life of public administration; the interesting thing here would be to discuss what such a procedure would mean for the different dimensions of resilience.

3 Urban resilience

In section 3 there are overlaps with respect to section 2, and, moreover, the potential readers are not guided in a way that a better understanding of resilience (of urban areas) can be achieved. Lot of information presented here (as well as in section 2) is not necessary for a review on resilience, but is supplementing the overall discussion on multiple urban "risks". As such, the overall text should be re-worked to better mirror the title and introduction, or, alternatively, the authors may wish to put their focus on urban risk and related challenges (which are not only related to resilience). I do not agree with the statement made in line 496 ("resilience can therefore be defined as the concept that studies urban systems", this is valid for many other approaches. Resilience, in contrast, seems to be a theoretical construct helping us to explain urban susceptibilities (e.g. to natural hazards) , again I kindly would like to refer to the above-mentioned recent textbook on the topic (of course, there are lots of other sources from different fellows, including those of Alexander, Cutter, Kelman, Kuhlicke, etc. – some of them even in NHESS).

[Figure]

4 Methods...

As the authors would like to present a review on resilience, it is not clear why in the methods section only methods for assessing (some) resilience indicators are presented, and not also matrices or even kind of equations of functions. As such, the overall manuscript seems to be targeted at (a) resilience indicators to measure (b) urban resilience. As such, the selection of Heinzlef et al.'s approach is not well explained. Futhermore, material presetend in section 4.2 is only loosely connected to the preceding (sub-)sections. Why we need DSS to measure resilience? Why do we need geo-visualisation? What do the authors want to tell us when presenting the DOMINO and the ViewExposed tools, and what are the differences to other available tools? The need for section 4.3 is, moreover, also not clear to me.

As such, the manuscript has some major weaknesses before we can conclude that it "has provided a review on the concept of resilience and its operationalization (cf. section 6). Consequently, it needs a re-writing over larger parts and a re-organisation before it may serve as a review paper on the term. Furthermore, key papers dealing with resilience in a multi-disciplinary context (and with respect to natural hazard risk management) are missing.These may not only include those originating in social sciences, but also in technical sciences and economics. Many (nearly all) sections are not very well connected so that the content of the latter section is prepared by certain gaps presented in the first one. Moreover, as stated above, I highly recommend to restrict the overall message to "urban planning" or "risk management in an urban context", also in the Abstract and in the Heading.

References mentioned

Emrich, C. T., and Tobin, G. A.: Resilience: An introduction, in: Vulnerability and resilience to natural hazards, edited by: Fuchs, S., and Thaler, T., Cambridge University Press, Cambridge, 124-144, 2018. Fuchs, S., and Thaler, T.: Vulnerability and resilience to natural hazards, Cambridge University Press, Cambridge, 336 pp., 2018.

International Standards Organisation: ISO 31000:2009, Risk management - Principles and guidelines, Geneva, pp., 2009. Papathoma-Köhle, M., Cristofari, G., Wenk, M., and Fuchs, S.: The importance of indicator weights for vulnerability indices and implications for decision making in disaster management, International Journal of Disaster Risk Reduction, 36, 101103, https://doi.org/10.1016/j.ijdrr.2019.101103, 2019.

---

## Author Comment (AC1) · 12 Sep 2020

**Answers to Reviewer**
**Resilience issues and challenges into built environments: a review**

**Reviewer 1:**

| Comments | Answers |
|---|---|
| Re-organise in the way of shortening this section due to large overlap with the Alexander DE (2013) wich published in the same journal. | We have deleted the part with the origin of the concept and the different disciplines that may have solicited it. And we added the recommended reference |
| Line 37-38: Rephrase as: "Despite its growing success, the op- erational relevance of the concept is therefore constantly being questioned." | Done |
| Line 42: "Over the past 20 years or so,. . ." Rephrase as "During last two decades" | Done |
| Line 62: The sentence too equivoke in it is meaning. "Interdisciplinary" itself, or "interdisciplinary ap- proach/ interdisciplinary studies" can serve it. Make clear please | We actually deleted this sentence as we reformulated this part (and cancel the origins of the concept) |
| Line 245: Rephrase as "Faced with increasing risks, stakeholders have identified two concepts (Saunders and Becker, 2015); resilience (taking into account the management of disturbances) and sustainable development (analyzing the balanced economic, social and environ- mental development of the territory)." | Done |
| Line 307: change "metres" with "meters". | Done |
| Line 415-416: Remove the sentence. You already mentioned in the previous sentence. An- other option: you can merge two sentences. | We deleted |
| Line 422: Replace "IC" with "CI" Line 428: Replace "metro" with "subway" | Done |
| Line 513: Replace subtitle of " A complex urban system. . ." with "Complexity of an urban system" | Done |
| Line 515: A short introductory sen- tence required. ". . .of this lack of clarity. . .." "this" refer what? | We added a sentence: "The diversity of definitions and subjects of analysis of urban resilience can be explained in particular by the complexity of current urban systems." |
| Line 520: "complicates", use capital letter when you start to a new sentence. | We entirely rebuilt the sentence: "Urban growth combined with urban, social, technical, political and economic changes leads to a fragmentation of urban space. This fragmentation and increasing complexity makes it difficult to build a |

| | shared knowledge on urban space, which is a prerequisite for adequate risk management." |
|---|---|
| Line 552: Replace subtitle of " . . . Including some limits" with "Some important issues of limitations of resilience concept" | Done |
| Line 554-555: Remove the whole sentence | Done |
| Line 560: Replace "in multiple fields" with "in many fields". | Done |
| Line 632: This paragraph not connected with its subtitle. It seems to be belong the next section. If so, move there | It was the transition but we made the change |
| Line 694: Remove "the Disaster Risk Index (UNDP 2004)." Connect "or" with "the Envi- ronmental Sustainability Index." | Done |
| Line 696: Two times you refer United States. Remove one of them. | Done |
| Line 702: Remove "." after "future" | Done |
| Line 727: Cutter et al. (2014) is better. | Done |
| Line 728-730: Gramatically incorrect sentence. Rearrange! | Done: ". Cutter divides resilience into six indicators: social, economic, community, institutional, infrastructural and environmental (Cutter et al., 2014)." |
| Line 746-47: Rewrite the whole sentence. A very difficult sentence to understand. | We did the change: "This is why the proposed work is done at the national scale and during a precise time scale without any comparative work over several years." |
| Line 817-819 Many "and" in one sentence. Rearrange! | We did the change: "When territorial issues are addressed, these are referred to as spatial decision Support System (DSS). They combine spatial with non-spatial data, functions analysis with visualization of Geographic Information Systems (GIS) and decisions in order to construct, evaluate and produce solutions (Keenan and Jankowski, 2019)." |
| Line 851: Remove "." | Done |
| Line 852: Remove "." | Done |
| Line 866: Critical Infrastructures (CI) already abbreviated in previous section. Use only the short form. | Done |
| Line 871: "it is" used two times | We deleted the second one |
| Line 884: "s"? | We deleted the "s" |
| Line 887 "territory of Norway" is better. | Done |
| Line 927: An Introductory sentence required for why you are going to set a series of subtitles. | We added an introductory sentence: "The objective is to build local actions and alliances to ensure that each actor understands his or her role in reducing and preparing risk reduction and resilience |

| | strategies (Heinzlef et al., 2020b; Gupta et al., 2010). Collaborative approaches are therefore essential levers in the process of involving, understanding and adopting the concept of resilience in risk management strategies by local stakeholders." |
|---|---|
| Line 953-988: You should start with introductory sentences. Then harmonize examples and/or approaches by citing references to present instead of give the summary of the study. | We added an introductory sentence: "There are several examples of collaborative and/or participatory approaches that aim to integrate local actors in the process of operationalizing resilience. We present two of them, whose case studies are Paris and Mexico City." |
| Line 977: You already cite to Freeman et al (2020). Is design methodology adopted from Brown et al?. If so, rephrase. If not, clarify it. | We added a precision: "Freeman et al. have developed a Resilience by Design methodology inspired by Brown et al., (2020)" |
| Line 991: No connection of the paragraph with the previous one and with its title. | We added this paragraph to the discussion |
| Line 992: Some references required. ". . ..risk strategies (reference/s) | We added references: "Several methodologies exist in order to operationalize resilience concepts and integrate it into urban risks strategies (Cutter et al., 2008; Heinzlef et al., 2020, 2019; Opach and Rød, 2013; Robert et al., 2008; Serre, 2018; Freeman et al., 2020; Toubin et al., 2015). » |
| Line 1005: In Discussion section it is expected from Authors to summarize basic arguments/findings in first paragraph after an introductory paragraph and continue with one or two more paragraph wich discuss "strong points" of the presented approach and "constrains". In this study you summarize and presented existing model in Table and give a figure about resilence observatory. I recommend to write this section under different title. In current form this is not "Discussion" of the presented study. | We deleted the title "discussion" and named it "From a multitude of operationalization methods to a resilience toolbox" |

**Reviewer 2:**

| Comments | Answers |
|---|---|
| The paper is very long (too long from my point of view) and could be substantially reduced without missing the main goal. | We have considerably reduced the manuscript by removing, according to the advice of the reviewers, unnecessary parts (origin of the concept of resilience, and |

| | some parts on the issue of increasing urbanization). |
|---|---|
| The paper deals with a social science concept and although some links to natural hazards are given, the question arises as to whether NHESS is the right target journal for this paper. I would have expected more information on how the concept could help to improve risk management of natural hazards. At the very end, the authors argue that a tool would be needed but it remains vague, what such a tool should contain and how such a tool should look like. May be that for a theoretical review of this kind of concepts other journals would be more appropriate | Regarding the suitability of the paper for the NHESS journal, this review was specifically requested by the editors. This review is part of a special issue of NHESS entitled "Resilience to risks in built environments". The editors have been asked to produce a review specific to the theme. This paper is therefore part of a specific research publication. |
| • line 37: What is the growing success? In the next line you write the concept is questioned. Isn't this a contradiction? | we have changed the term "success" to "use in official communications".

It is contradictory, an increasing use of the term but a lack of operationalization. This is what is at stake in the subject and in the debate. |
| • line 49: at the end of the sentence, I think a reference would be needed. | We deleted this paragraph in order to shorten the paper, regarding reviewer 1 advices. |
| • line 132: why "attempted resilience definitions"? Consider to change the section title. | We changed for "many definitions of resilience" |
| • line 134: … many different disciplines … redundant to paragraph above. | As we have completely reworded the first paragraph, we maintain this sentence which no longer has the same meaning of repetition |
| line 136: you probably mean the criteria for determining when a system has recovered. | Yes, exactly.

We added a precision: "when and according to what criteria can it be determined that a system has recovered from the number of disturbances, changes and transformations it has undergone?" |

| | |
|---|---|
| lines 147 – 157: check for redundancies and language. Which capacities did Serre (2018) mean? | We added a precision: "Serre (2018) defined three capacities (resistance, absorption, recovery) of resilience and defined the resistance ability to determine "*the physical damage to the network as a result of the hazard*" (Serre et al., 2013)." |
| lines 218 – 219 very unspecific. What is positive? | We added precisions: "Vulnerability of a system is seen as a positive element when it leads to change that results in beneficial transformation". |
| lines 236 – 237: I do not understand what you want to say here. | We reformulated: "Both concepts are equally suitable for the analysis of technical and/or social systems." |
| lines 277 – 278: "The concept . . . " this has been said several times. | We canceled the formulation: "Resilience is a multifaceted concept, involving a plurality of disciplines, definitions, notions and associated concepts" |
| lines 291 – 292 what exactly has increased? The number? | Yes: "The current climate change context has led to an increase in the number of natural disasters of about 2% per year for the past 15 years" |
| lines 303 – 309: where do these numbers come from? (UN, 2018)? | Yes, we added this reference at the end. |
| line 322: isn't it primarily the exposure that increased rather than the vulnerabil- ity? Of course both could have increased, depends on the definition. | we had already mentioned it above (link between exposure and vulnerability) but we have added it again: "Thus, between 1946 and 2007, urbanization , and therefore exposure, in the lower areas doubled or even tripled in some communes, leading to significant vulnerability." |
| lines 328 – 330: ...live discontinuously from ...natural functioning ...natural functioning: very general | Yes but it is a key element. Knowledge of the territory and collective memory are key elements for the implementation of resilience but also for an adequate and relevant risk management strategy. |
| lines 335 – 337: repetitions, consider rephrasing; also very general. | We cancelled a sentence to summarize. |
| lines 339 – 364: is the description here really necessary? Same holds for lines 370 – 400; could be at least reduced. | We summarized and cancel several sentences and details. |
| lines 401 – 489: the question would be here, how the concept of resilience could help to better deal with these complex systems. You mainly describe the com- plexity. I suggest to | We cut these paragraphs. |

| | |
|---|---|
| cut these sections to closer link it to resilience. | |
| line 499 – 500: is urban resilience really a tool? Shouldn't a resilience tool be rather a tool for analyzing the complexity and ways to improve it. In this context (and for the whole paper): what the difference between resilience and robust- ness? I think this term should be mentioned and defined somewhere. | We added a precision: "Urban resilience would therefore be a utilitarian concept for analyzing the complexity of the urban system and defining the different capacities and capabilities of each element that defines this system in order to live and survive a disruptive event."

Robustness is only one element among many that can be integrated into the notion of resilience. |
| table 1: citations should be consequently added in the left column. | it was already the quotes. We added the quotation marks |
| line 513 and 552: check the section titles and rephrase | We made changes in the titles

3.2. Complexity of an urban system

3.3. Some important issues of limitations of resilient concept |
| line 518 – 519: check the sentence | We reformulated |
| line 539: difference between city and town? | We cancel this part of the sentence |
| lines 559 – 561: this sentence better fits at the beginning and is repeated several times | We reformulated: "As presented in the introduction, resilience concept is over-used, over-solicited in many fields and related to several concepts (Emrich and Tobin, 2018)." |
| I think this part is the most essential one, while the upper sections could be shortened as much as possible. I recommend to reduce general descriptions such the one on "indicators" (lines 669 –680) as much as possible. | we have reduced the indicators part. |
| • ines 813: Check sentence, same next line: "taking" is a complex ...not clear what you mean. | We reformulated: "The need to create decision support systems is logical given the abstraction of the concept. In risk management, decision making is a complex combination of knowledge management and |

| | |
|---|---|
| | decision-making processes (Tacnet et al., 2014)." |
| • Thesection4.2.2isverygenerallyandcouldbeshortenedtothoseaspectsreally relevant here | We have reduced this part |
| • line 871: ...it is It is ... | We delete it |
| • line 884: . . . territories s response | We delete it |
| • line 886: . . . including . . . please check. | We made a change "the aim of which is to inform local authorities" |
| I think this is not really a discussion. I would expect that you discuss which of the concepts are used in practice and when not (what I assume for most of them), what's the reason for that. Is it the concept itself or the way it is implemented? What is missing that the concept could be used and implemented in practice? Wouldn't figure 1 better fit in one of the upper chapters? | Following the advice of reviewer 1, we have modified what was previously the section entitled "discussion". We have transformed our comments as well as the title. Thus this part, which allows us to open up innovative perspectives is called: " from a multitude of operationalization methods to a resilience toolbox" |
| • multiplication → multiplicity | Done |
| • ...perfectly in line ...you perhaps mean ...is able to be used in the context of climate change and . . . Next line: currently challenge → current challenge | We changed: "Yet this concept, which encourages adaptability, evolution and flexibility, is perfectly adequate for the analysis of climate change and the associated risks and uncertainties" Done |
| • The last sentence sounds very hypothetical and you could say more in the discussion how this tool could look like. | Given that this project is under development (as presented and specified above), the perspective of finality is indeed a scientific hypothesis that needs to be justified in the coming months. |
| • line 13/14: . . . innovate existing ? risk management strategies • | We added a precision: ". In a context of climate change, increased risks in urban areas and growing uncertainties, urban managers are forced to innovate in order to design appropriate new risk management strategies." |
| • line 34: ...such as physics ... • | Done |

| | |
|---|---|
| • line 54: resiliences | It is with a "s" as explained, because the concept has several definitions and meanings. |
| • line 123: word identity double | Done |
| • line 142: We evaluate . . . | We changed: "In this case, we analyze the resilience capacity of a post-crisis system (outcome), or the succession of solutions developed by this system to recover from a shock (process)." |

**Reviewer 3:**

| Comments | Answers |
|---|---|
| The introductory paragraph (1) should focus more clearly on the overall paper aim (resilience and related challenges for the built environment) since operationalising re-silience is a challenging issue with different disciplinary roots. | We added the precision in built environments: "Operationalizing urban resilience is a complex, even conflicting subject. Because of its multidisciplinary origin and the multitude of approaches, interpretations of resilience and its operationalization are sometimes contradictory (Davoudi et al., 2012)." … "The concept of resilience is faced with a problem of formalization which makes it difficult to go beyond the purely theoretical use of the concept in order to promote its concrete and useful use for urban actors and managers (Weichselgartner and Kelman, 2015)." |
| Moreover, "risk management" is not a discipline but a method used by multiple disciplines! | We cancelled it and focused on these disciplines: "This contradiction is essentially due to the fact that resilience belongs to many disciplines such as physics, psychology, ecology" |
| Moreover, the authors further argue that this "disciplinary and conceptual vagueness makes the use f resilience and its integration into risks" – which is neither grammatically clear, nor from a subject point of view (in particular because this "vagueness" has not been introduced before). | We rephrased it : "This disciplinary and conceptual vagueness makes the use of resilience and its integration into risk management complex" |

| | |
|---|---|
| Further, it is not clear why it is challenging to "move from theory to practice". | We rephrased our purpose: "The concept of resilience is faced with a problem of formalization which makes it difficult to go beyond the purely theoretical use of the concept in order to promote its concrete and useful use for urban actors and managers (Weichselgartner and Kelman, 2015)." |
| Again, this section starts with some strong statements which I cannot follow. While vulnerability is an integral part of risk management (e.g., International Standards Or- ganisation, 2009), resilience is only if defined as the counterpart of vulnerability | This is not the point of view defended by this article nor by the authors' research. As explained in 1.4.1 Resilience vs. Vulnerability, the two concepts have historically been opposed, one being the counterpart of the other. However, much research in the field of risk management has contradicted this view. For example, and this is what is presented here, if resilience is considered as a process, this concept and that of vulnerability would be placed on the same continuum (Manyena, 2006). Moreover, the opposition between the two concepts is based primarily on two critical acceptances. First, resilience is a positive element of a system that needs to be increased, while vulnerability is a negative element that needs to be decreased (Pelling, 2003). Second, resilience and vulnerability would be the opposite of each other, they would be two opposite sides of the same coin (Folke et al., 2002). These two hypotheses are questionable. Indeed, the negative aspects of resilience are established and not all resilience is "good to take" (Ruffat, 2010; Reghezza et al., 2012). The second postulate would lead to a tautological reasoning consisting of wanting to reduce vulnerability in order to increase resilience and conversely wanting to increase resilience in order to reduce vulnerability (Klein et al., 2003).

 Folke C. et al., 2002, Resilience and Sustainable Development: Building Adaptive Capacity in a World of Transformations, Environmental Advisory Council to the Swedish Government, Stockholm, Sweden.

 Klein R. J., Nicholls R. J., Thomalla F., 2003, « Resilience to Natural Hazards: How Useful is the Concept? », Environmental Hazards, Vol. 5, n°1-2, pp. 35-45.

 Manyena S. B., 2006, « The concept of resilience revisited », Disasters, 30(4), pp. 434-450.

 Pelling M., 2003, The Vulnerability of Cities: social resilience and natural disaster, Earthscan, London.

 Reghezza M., Rufat S., Djament-Tran G., Leblanc A., Lhomme S., 2012, « What resilience is not : Resilience use and abuse », Cybergeo.

 Rufat S., 2010, « Bucarest entre inertie et resilience, perennite urbaine », in traces, ed. Harmattant, pp. 92-101. |
| As such, the introduction to section 1.1 needs careful revision, also with respect to the overall disciplinary use of the | point 1.1. has been reshaped following the advice of the other two reviewers (shortcut).

 We have not exactly presented resilience in a global manner but have dissected it according to different disciplines such as |

| | |
|---|---|
| term resilience and related conceptualization in risk management. The same is valid for the subsequent sentence stating that the concept of resilience is "over-used" | physics, psychology or ecology. This part was reduced according to the advice of the reviewers.

However, we maintain the assertion that resilience in risk management is a disciplinary tool. Let us also recall the Foucauldian perspective that sees the discipline as a technology, which allows us to define risk management as a disciplinary technology with tools.

Concerning the term "over-used", we also maintain it. The term is now used everywhere, from political speeches (Paris résilient, https://www.paris.fr/pages/paris-resiliente-4264 ), to entrepreneurial injunctions (https://hbr.org/2020/07/a-guide-to-building-a-more-resilient-business ) or as a tool to prepare territories and populations for increased risks in a climatic context (100 Resilient Cities, n.d.; Chelleri, 2012; Mendizabal et al., 2018; Resilient Vejle and 100 Resilient Cities, 2013; UNISDR, 2012)

100 Resilient Cities, n.d. The City Resilience Framework.
Chelleri, L., 2012. From the «Resilient City» to Urban Resilience. A review essay on understanding and integrating the resilience perspective for urban systems. Doc. Anàlisi Geogràfica 58, 287. https://doi.org/10.5565/rev/dag.175
Mendizabal, M., Heidrich, O., Feliu, E., García-Blanco, G., Mendizabal, A., 2018. Stimulating urban transition and transformation to achieve sustainable and resilient cities. Renew. Sustain. Energy Rev. 94, 410–418. https://doi.org/10.1016/j.rser.2018.06.003
Resilient Vejle, 100 Resilient Cities, 2013. Vejle's resilience strategy. 100 Resilient Cities.
UNISDR, 2012. Making Cities Resilient. The United Nations Office for Disaster Risk Reduction. |
| In my opinion the overall introduction to section 1 should also explain why the sub- sequent sections are focusing solely on physics, psychology, ecology and "risk man-agement", and not e.g. also on social sciences other than psychology or economics. Moreover, the question is if we could use the ecological concept of resilience to explain observations in natural hazard risk management, such as e.g. the idea of "building back better" (see e.g. | This part has, as explained above, been shortened considerably since it was not the core of the article.

In addition, the reason why we have further detailed disciplines such as physics, psychology or ecology is that some of the characteristics of the definitions of resilience in these disciplines are found in risk management. This is due to a logical and historical evolution and construction. If we take the example of ecology, it is because the systems approach of an ecosystem has been adopted for the analysis of the resilience of urban systems. If we take the example of psychology, the characteristics of resilience can be found in the approach to the resilience of societies in the face of a disaster, etc.

We added: "It can also be understood from the ecological angle of "building back better" (Papathoma-Köhle et al., 2019)" |

| | |
|---|---|
| discussion in Papathoma-Köhle et al. 2019). | |

| | We added a table : |
|---|---|
| Here it would be nice to see a kind of table to better shown contradicting and simi- lar characteristics of conceptualizing resilience, | |

| Capacities | Definitions | Contradictory | Complementary |
|---|---|---|---|
| **Resistance** | ability to determine "the physical damage to the network as a result of the hazard" | Resistance / absorption | |
| **Absorption** | to absorb negative impacts and recover from these | Absorption / Resistance | Absorption + Adaptive+ Learning |
| **Adaptive** | "ability of systems, institutions, humans and other organisms to adjust to potential damage, to take advantage of opportunities, or to respond to consequences" | | Adaptive + Reaction + absorption + learning |
| **Reaction** | a "capacity of systems to reorganize and recover from change and disturbance". | | Reaction + Adaptive |
| **To rebuild** | to "reorganize while undergoing change so as to still retain essentially the same function, structure identity, and feedbacks | | Rebuild + Bounce back |
| **Learning** | the degree to which the system can | Learning/ Bounce back | Learning + absorption + |

| | | | | |
|---|---|---|---|---|
| | | build and increase the capacity for learning and adaptation | | adaptive + reaction |
| | **To bounce back** | equilibrium which implies to bounce back to equilibrium previous disturbance | Bounce Back/Learning | Bounce back + rebuild |

| As this manuscript should provide an overview on different pathways to resilience I am wondering why only one perspective (the one of social sciences) is taken in section 1.3.1 – it is clear that different disciplinary foci exist, but a review should provide an overview on the main concepts and as such I am missing at least a "physical" and "economic" approach here, and these can co-exist together with the socioscientific approach of seeing resilience and vulnerability not as counterparts but as additives in risk management (as vaguely stated in lines 235 ff.). This should be more elaborated. | Since 1.2. we focus on resilience in risk management and no longer in other areas. We have therefore focused our argument on this discipline and cannot develop resilience in Physics or Economics. However, the term social sciences needs to be nuanced since risk management (and particularly with Toubin's references) also integrates engineering approaches. |
|---|---|
| As far as I understood the overall manuscript is centered on urban areas, as such this should be better reflected in the title (instead of "built environments"). Moreover, at least from section 2 onwards we would need a proper definition of how risk and vulnerability are | We prefer to keep this title because it makes a direct link with the title of the special issue.

Concerning the definition that would be specific to the authors, this does not seem relevant to us since the article is a review and therefore does not represent a personal point of view. Our review objective is not to propose a single definition of resilience but to present the range of definitions, points of view and applications of the concept of resilience in risk management. |

| | |
|---|---|
| understood by the authors so that the overall aim of providing a review on the ruse of different types of resilience can be better understood with respect to and opposed to the term "resilience" and the specific use with respect to urban areas (?). Sections 2 and 3 are then a bit abruptly focusing on critical infrastructure in urban areas, if this is the overall aim here also CI and even networks as part of CI should be mentioned earlier and mirrored in the title accordingly. Otherwise, the sections and paragraphs need to be better connected so that potential readers will be guided through the use of the term "resilience" in urban areas and with respect to critical infrastructure in cities or even networks. | We have added transitions: "Resilience in risk management is particularly relevant in addressing the increased vulnerability of urban areas. Urban areas are in fact the territories most exposed to disasters. The current climate change context has led to an increase in the number of natural disasters of about 2% per year for the past 15 years (Catastrophes Naturelles-Observatoire permanent des catastrophes naturelles et des risques naturels, 2016). At the same time, the increase in the number of people and goods in urban areas is making it more fragile. considerably the cities. Today, nearly three out of five cities, with 500,000 inhabitants, are at risk. However, urban areas produce between 70 and 80% of the world economy and are home to 55% of the world's population , with an increasing urban-rural drift expected to raise this value up to 68% by 2050 (UNDESA, 2019; Zevenbergen et al., 2010). Such a concentration of stakes increases the impact of disasters (Boin and McConnell, 2007). and raises questions on the future of cities."

We have given the example of critical infrastructure (shortened section) because it demonstrates the complexity of urban areas and their over-vulnerability and therefore the need to implement urban resilience strategies. |
| It remains unclear why the discussion on integrating multi-(hazard and) risk in the man-agement of urban areas is necessary for the review on the resilience term. | we have deleted this subpart |
| In section 3 there are overlaps with respect to section 2, and, moreover, the potential readers are not guided in a way that a better understanding of resilience (of urban areas) can be achieved. Lot of information presented here (as well as in section 2) is not necessary for a review on resilience, but | We have reduced part 3 to avoid redundancy with part 2.

We would like to make it clear that we do not define resilience as follows: "resilience can therefore be defined as the concept that studies urban systems".

Our definition is as follows: "Urban resilience can therefore be defined as the concept that studies urban systems faced to urban risks, i.e. the interactions between the different components that participate in the creation of the territory and that can be impacted by the risks.. Urban resilience refers to a systemic approach that encompasses the multiple layers (built, social, political, etc.) and |

| | |
|---|---|
| is supplementing the overall discussion on multiple urban "risks". As such, the overall text should be re-worked to better mirror the title and introduction, or, alternatively, the authors may wish to put their focus on urban risk and related challenges (which are not only related to resilience). I do not agree with the statement made in line 496 ("resilience can therefore be defined as the concept that studies urban systems", this is valid for many other approaches. Resilience, in contrast, seems to be a theoretical construct helping us to explain urban susceptibilities (e.g. to natural hazards) , again I kindly would like to refer to the above-mentioned recent textbook on the topic (of course, there are lots of other sources from different fellows, including those of Alexander, Cutter, Kelman, Kuhlicke, etc. – some of them even in NHESS). | structures that produce an integrated vision of the urban object. Urban resilience would therefore be a utilitarian concept for analyzing the complexity of the urban system and defining the different capacities and capabilities of each element that defines this system in order to live and survive a disruptive event." |
| As the authors would like to present a review on resilience, it is not clear why in the methods section only methods for assessing (some) resilience indicators are pre- sented, and not also matrices or even kind of equations of functions. As such, the overall manuscript seems to be targeted at (a) resilience indicators to measure (b) urban resilience. As such, the selection of Heinzlef et al.'s approach is not well ex- plained. Futhermore, material presetend in | Concerning the method part, as explained, given the diversity of approaches and models seeking to operationalize resilience, we had to make categories: indicators to measure urban resilience (which is not only defined by an urban structure, as explained by Heinzlef, but also by an urban population), geovisualization techniques (in order to visualize the mapping of resilience measurement), and collaborative approaches (in order to ensure the understanding and adoption over time of urban resilience strategies by local actors).

Concerning the link between point 4.2 "Modelling resilience", the link has been defined by the introductory sub-sections 4.2.1 and 4.2.2 to justify the use of spatial decision support systems in the operationalization of resilience: "As the concept of resilience is a complex subject to address and operationalize for local actors, many tools have been created to |

| | |
|---|---|
| section 4.2 is only loosely connected to the preceding (sub-)sections. Why we need DSS to measure resilience? Why do we need geo-visualisation? What do the authors want to tell us when presenting the DOMINO and the ViewExposed tools, and what are the differences to other available tools? The need for section 4.3 is, moreover, also not clear to me. | simplify, define, measure and attempt to operationalize this concept. The need to create decision support systems is logical given the abstraction of the concept. In risk management, decision making is a complex combination of knowledge management and decision-making processes (Tacnet et al., 2014)." And "Geovisualization thus integral part of spatial decision support systems, as it allows to meet both scientific and societal needs to initiate a process of reflection and thereby build and produce knowledge.Several methodologies have produced tools to clarify the concepts of resilience and vulnerability. These tools are spatial decision support systems and have made it possible to dissect the concept of resilience. The objective of each of these approaches is to make the concept accessible by creating links between scientific advances and territorial reality."

Finally, to justify our choice of presented methodologies, we are obviously aware of the non-exhaustiveness of the presentation. Nevertheless, the models presented are among the only ones that have been designed by scientific experts and are currently used by local actors and managers. Contrary to some of them, which were designed only by international institutions or groups, or others which were developed by scientists but not adopted by local actors, the models presented are scientifically constructed AND operational at the local level. |
| As such, the manuscript has some major weaknesses before we can conclude that it "has provided a review on the concept of resilience and its operationalization (cf. section 6). Consequently, it needs a re-writing over larger parts and a re-organisation before it may serve as a review paper on the term. Furthermore, key papers dealing with resilience in a multi-disciplinary context (and with respect to natural hazard risk management) are missing.These may not only include those originating in social sci-ences, but also in technical sciences and economics. Many (nearly all) sections are not very well connected so that the | We have taken your comments into account as we go along (as well as those of the other reviewers) and we therefore respond to all your remarks. |

| content of the latter section is prepared by certain gaps presented in the first one. Moreover, as stated above, I highly recommend to restrict the overall message to "urban planning" or "risk management in an urban context", also in the Abstract and in the Heading. | |
| --- | --- |